# Derandomized novelty detection with FDR control via conformal e-values

**Meshi Bashari**
Department of Electrical and Computer Engineering
Technion IIT
Haifa, Israel
meshi.b@campus.technion.ac.il

**Amir Epstein**
Citi Innovation Lab
Tel Aviv, Israel
amir.epstein@citi.com

**Yaniv Romano**
Department of Electrical and Computer Engineering
Department of Computer Science
Technion IIT
Haifa, Israel
yromano@technion.ac.il

**Matteo Sesia**
Department of Data Sciences and Operations
University of Southern California
Los Angeles, California, USA
sesia@marshall.usc.edu

## Abstract

Conformal inference provides a general distribution-free method to rigorously calibrate the output of any machine learning algorithm for novelty detection. While this approach has many strengths, it has the limitation of being randomized, in the sense that it may lead to different results when analyzing twice the same data, and this can hinder the interpretation of any findings. We propose to make conformal inferences more stable by leveraging suitable conformal *e-values* instead of *p-values* to quantify statistical significance. This solution allows the evidence gathered from multiple analyses of the same data to be aggregated effectively while provably controlling the false discovery rate. Further, we show that the proposed method can reduce randomness without much loss of power compared to standard conformal inference, partly thanks to an innovative way of weighting conformal e-values based on additional side information carefully extracted from the same data. Simulations with synthetic and real data confirm this solution can be effective at eliminating random noise in the inferences obtained with state-of-the-art alternative techniques, sometimes also leading to higher power.

## 1 Introduction

### 1.1 Background and motivation

A common problem in statistics and machine learning is to determine which samples, among a collection of new observations, were drawn from the same distribution as a reference data set (Wilks, 1963; Riani et al., 2009; Chandola et al., 2009). This task is known as *novelty detection, out-of-distribution testing, or testing for outliers*, and it arises in numerous applications within science,

37th Conference on Neural Information Processing Systems (NeurIPS 2023).

engineering, and business, including for example in the context of medical diagnostics (Tarassenko et al., 1995), security monitoring (Zhang et al., 2013), and fraud detection (Ahmed et al., 2016). This paper looks at the problem from a model-free perspective, in the sense that it does not rely on parametric assumptions about the data-generating distributions, which are generally unknown and complex. Instead, we apply powerful machine learning models for one-class (Moya et al., 1993) or binary classification to score the new samples based on how they *conform* to patterns observed in the reference data, and then we translate such scores into rigorous tests using conformal inference.

Conformal inference (Vladimir et al., 2005; Lei et al., 2013) provides flexible tools for extracting provably valid novelty detection tests from any *black-box* model. The simplest implementation is based on random sample splitting. This consists of training a classifier on a subset of the reference data, and then ranking the output score for each test point against the corresponding scores evaluated out-of-sample for the hold-out reference data. As the latter do not contain outliers, the aforementioned rank is uniformly distributed under the null hypothesis that the test point is not an outlier (Laxhammar and Falkman, 2015; Smith et al., 2015; Guan and Tibshirani, 2022), as long as some relatively mild *exchangeability* assumptions hold. In other words, this calibration procedure yields a conformal *p-value* that can be utilized to test for outliers while rigorously controlling the probability of making a *false discovery*—incorrectly labeling an *inlier* data point as an "outlier". Further, split-conformal inference produces only weakly dependent p-values for different test points (Bates et al., 2023), allowing exact control of the expected proportion of false discoveries—the *false discovery rate* (FDR)—with the powerful Benjamini-Hochberg (BH) filter (Benjamini and Hochberg, 1995).

As visualized in Figure 1a, a limitation of split-conformal inference is that it is randomized—its results for a given data set are unpredictable because they depend on how the reference samples are divided between the training and calibration subsets. However, higher stability is desirable in practice, as randomized methods generally tend to be less reliable and more difficult to interpret (Murdoch et al., 2019; Yu and Kumbier, 2020). This paper addresses the problem of making conformal inferences more stable by developing a principled method to powerfully aggregate tests for outliers obtained with repeated splits of the same data set, while retaining provable control of the FDR. This problem is challenging because dependent p-values for the same hypothesis are difficult to aggregate without incurring into a significant loss of power (Vovk and Wang, 2020; Vovk et al., 2022).

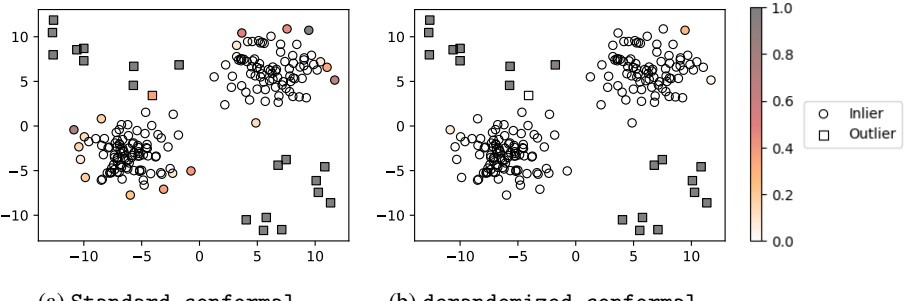

(a) `Standard conformal.`      (b) `derandomized conformal.`

Figure 1: Demonstration on two-dimensional synthetic data of standard conformal (a) and derandomized conformal (b) inferences for novelty detection. Circles denote true inliers and squares denote outliers. The colors indicate how often each test point is reported as a possible outlier over 100 independent analyses of the same data. By carefully aggregating evidence from 10 distinct analyses based on independent splits of the same data, the proposed derandomized approach discovers the same outliers consistently and is less likely to make random false discoveries.

## 1.2 Main contributions

This paper utilizes carefully constructed conformal *e-values* (Vovk and Wang, 2021) instead of *p-values* to quantify statistical significance when testing for outliers under FDR control. The advantage of e-values is that they make it possible to aggregate the results of mutually dependent tests in a relatively simple way, enabling an effective approach to derandomize conformal inferences. Our contribution is to develop a martingale-based method inspired by Ren and Barber (2023) that leverages e-value ideas efficiently, as different types of e-values can be constructed but not all would be powerful

in our context due to the discrete nature of the statistical evidence in conformal inference. We further refine this method and boost power by adaptively weighting our conformal e-values based on an estimate of the out-of-sample accuracy of each underlying machine learning model. A preview of the performance of our solution is given by Figure 1b, which shows that our method can achieve power comparable to that of standard conformal inferences while mitigating the algorithmic randomness.

### 1.3 Related work

This paper builds upon *e-values* (Vovk and Wang, 2021): quantitative measures of statistical evidence, alternative to p-values, that lend themselves well to the derandomization of data-splitting procedures and to FDR control under dependence (Wang and Ramdas, 2022). There exist several generic methods for converting any p-value into an e-value (Vovk and Wang, 2021). While those *p-to-e calibrators* could be applied for our novelty detection problem, their power turns out to be often quite low due to the fact that conformal p-values are discrete and cannot take very small values unless the sample size is extremely large; see the Supplementary Section S5 for more details.

Therefore, we propose a novel construction of (slightly generalized) e-values inspired by the work of Ren and Barber (2023) on the derandomization of the knockoff filter (Barber and Candès, 2015), which focused on a completely different high-dimensional variable selection problem. A different approach for producing e-values in the context of conformal inference can also be found in Ignatiadis et al. (2023), although the latter did not focus on derandomization. Our approach differs from that of Ignatiadis et al. (2023) because we construct e-values simultaneously for the whole test set, aiming to control the FDR, instead of operating one test point at a time. Simulations show that our approach tends to yield higher power, especially if the test data contain many outliers.

Our second novelty consists of developing a principled method for assigning data-driven weights to conformal e-values obtained from different machine learning models, in such a way as to further boost power. This solution re-purposes *transductive* (Vovk, 2013) conformal inference ideas to leverage information contained in the test data themselves while calibrating the conformal inferences, increasing the power to detect outliers similarly to Marandon et al. (2022) and Liang et al. (2022).

While this paper focuses on derandomizing split-conformal inferences, there exist other distribution-free methods that can provide finite-sample tests for novelty detection, such as full-conformal inference (Vladimir et al., 2005) and cross-validation+ (Barber et al., 2021). Those techniques are more computationally expensive but have the advantage of yielding relatively more stable conformal p-values because they do not rely on a single random data split. However, full-conformal inference and cross-validation+ also produce conformal p-values with more complicated dependencies, which make it difficult to control the FDR without large losses in power (Benjamini and Yekutieli, 2001) or very expensive computations (Fithian and Lei, 2022; Liang et al., 2022).

Finally, prior works studied how to stabilize conformal predictors by calibrating the output of an *ensemble* of simpler models (Löfström et al., 2013; Beganovic and Smirnov, 2018; Linusson et al., 2020; Kim et al., 2020; Gupta et al., 2022). However, we consider a distinct problem as we focus on derandomizing conformal novelty detection methods while controlling the FDR.

## 2 Relevant technical background

### 2.1 Notation and problem setup

Consider $n$ observations, $X_i \in \mathbb{R}^d$, sampled exchangeably (or, for simplicity, independent and identically distributed) from some unknown distribution $P_0$, for all $i \in \mathcal{D} = [n] = \{1, \ldots, n\}$. Then, imagine observing a test set of $n_{\text{test}}$ "unlabeled" samples $X_j \in \mathbb{R}^d$. The problem is to test, for each $j \in \mathcal{D}_{\text{test}} = [n + n_{\text{test}}] \setminus [n]$, the *null hypothesis* that $X_j$ is also an *inlier*, in the sense that it was randomly sampled from $P_0$ exchangeably with the data in $\mathcal{D}$. We refer to a rejection of this null hypothesis as the *discovery* that $X_j$ is an *outlier*, and we indicate the set of true inlier test points as $\mathcal{D}_{\text{test}}^{\text{null}}$, with $n_{\text{test}}^{\text{null}} = |\mathcal{D}_{\text{test}}^{\text{null}}|$. For each $j \in \mathcal{D}_{\text{test}}$, define $R_j$ as the binary indicator of whether $X_j$ is labeled by our method as an outlier. Then, the goal is to discover as many true outliers as possible while controlling the FDR, defined as $\text{FDR} = \mathbb{E}[(\sum_{j \in \mathcal{D}_{\text{test}}^{\text{null}}} R_j) / \max\{1, \sum_{j \in \mathcal{D}_{\text{test}}} R_j\}]$.

## 2.2 Review of FDR control with conformal p-values

After randomly partitioning $\mathcal{D}$ into two disjoint subsets $\mathcal{D}_{\text{train}}$ and $\mathcal{D}_{\text{cal}}$, of cardinality $n_{\text{train}}$ and $n_{\text{cal}} = n - n_{\text{train}}$ respectively, the standard approach for computing split-conformal p-values begins by training a one-class classification model on the data indexed by $\mathcal{D}_{\text{train}}$. This model is applied out-of-sample to compute conformity scores $\hat{S}_i$ and $\hat{S}_j$ for all calibration and test points $i \in \mathcal{D}_{\text{cal}}$ and $j \in \mathcal{D}_{\text{test}}$, with the convention that larger scores suggest evidence of an outlier. Assuming without loss of generality that all scores take distinct values (otherwise, ties can be broken at random by adding a little noise), a conformal p-value $\hat{u}(X_j)$ for each $j \in \mathcal{D}_{\text{test}}$ is then calculated by taking the relative rank of $\hat{S}_j$ among the $\hat{S}_i$ for all $i \in \mathcal{D}_{\text{cal}}$: $\hat{u}(X_j) = (1 + \sum_{i \in \mathcal{D}_{\text{cal}}} \mathbb{I}\{\hat{S}_j \le \hat{S}_i\})/(1 + n_{\text{cal}})$. If the null hypothesis for $X_j$ is true, $\hat{S}_j$ is exchangeable with $\hat{S}_i$ for all $i \in \mathcal{D}_{\text{cal}}$, and $\hat{u}(X_j)$ is uniformly distributed on $\{1/(1 + n_{\text{cal}}), 2/(1 + n_{\text{cal}}), \ldots, 1\}$. Since this distribution is stochastically larger than the continuous uniform distribution on $[0, 1]$, one can say that $\hat{u}(X_j)$ is a valid conformal p-value. Note however that the p-values $\hat{u}(X_j)$ and $\hat{u}(X_{j'})$ for two different test points $j, j' \in \mathcal{D}_{\text{test}}$ are not independent of one another, even conditional on $\mathcal{D}_{\text{train}}$, because they share the same calibration data.

Despite their mutual dependence, conformal p-values can be utilized within the BH filter to simultaneously probe the $n_{\text{test}}$ hypotheses for all test points while controlling the FDR. A convenient way to explain the BH filter is as follows (Storey, 2002). Imagine rejecting the null hypothesis for all test points $j$ with $\hat{u}(X_j) \le s$, for some threshold $s \in [0, 1]$. By monotonicity of $\hat{u}(X_j)$, this amounts to rejecting the null hypothesis for all test points $j$ with $\hat{S}_j \ge t$, for some appropriate threshold $t \in \mathbb{R}$. An intuitive estimate of the proportion of false discoveries incurred by this rule is:

$$\widehat{\text{FDP}}(t) = \frac{n_{\text{test}}}{1 + n_{\text{cal}}} \cdot \frac{1 + \sum_{i \in \mathcal{D}_{\text{cal}}} \mathbb{I}\{\hat{S}_i \ge t\}}{\sum_{j \in \mathcal{D}_{\text{test}}} \mathbb{I}\{\hat{S}_j \ge t\}}. \tag{1}$$

This can be understood by noting that $\sum_{j \in \mathcal{D}_{\text{test}}} \mathbb{I}\{S_j^{(k)} \ge t\}$ is the total number of discoveries, while the numerator should behave similarly to the (latent) number of false discoveries in $\mathcal{D}_{\text{test}}$ due to the exchangeability of $\hat{S}_i$ and $\hat{S}_j$ under the null hypothesis. With this notation, it can be shown that the BH filter applied at level $\alpha \in (0, 1)$ computes an adaptive threshold

$$\hat{t}^{\text{BH}} = \min\left\{t \in \{\hat{S}_i\}_{i \in \mathcal{D}_{\text{cal}} \cup \mathcal{D}_{\text{test}}} : \widehat{\text{FDP}}(t) \le \alpha\right\}, \tag{2}$$

and rejects all null hypotheses $j$ with $\hat{S}_j \ge \hat{t}^{\text{BH}}$; see Rava et al. (2021) for a derivation of this connection. This procedure was proved by Bates et al. (2023) to control the FDR below $\alpha$.

## 2.3 Review of FDR control with `AdaDetect`

Recently, Marandon et al. (2022) proposed `AdaDetect`, a more sophisticated version of the method reviewed in Section 2.2. The main innovation of `AdaDetect` is that it leverages a binary classification model instead of a one-class classifier. In particular, `AdaDetect` trains a binary classifier to distinguish the inlier data in $\mathcal{D}_{\text{train}}$ from the mixture of inliers and outliers contained in the union of $\mathcal{D}_{\text{cal}}$ and $\mathcal{D}_{\text{test}}$. The key idea to achieve FDR control is that the training process should remain invariant to permutations of the calibration and test samples. While the true inlier or outlier nature of the observations in $\mathcal{D}_{\text{test}}$ is obviously unknown at training time, `AdaDetect` can still extract some useful information from the test data which would otherwise be ignored by the more traditional split-conformal approach reviewed in Section 2.2. In particular, `AdaDetect` can leverage the test data to automatically tune any desired model hyper-parameters in order to approximately maximize the number of discoveries. A similar idea also motivates the alternative method of *integrative* conformal p-values proposed by Liang et al. (2022), although the latter requires the additional assumption that some labeled outlier data are available, and is therefore not discussed in equal detail within this paper.

Despite a more sophisticated use of the available data compared to the split-conformal method reviewed in Section 2.2, `AdaDetect` still suffers from the same limitation that it must calibrate its inferences based on a single random data subset $\mathcal{D}_{\text{cal}}$, and thus its results remain aleatory. For simplicity, Section 3.1 begins by explaining how to derandomize standard split-conformal inferences; then, the proposed method will be easily extended in Section 3.3 to derandomize `AdaDetect`.

# 3 Method

## 3.1 Derandomizing split-conformal inferences

Consider $K \geq 1$ repetitions of the split-conformal analysis reviewed in Section 2.2, each starting with an independent split of the same reference data into $\mathcal{D}_{\text{train}}^{(k)}$ and $\mathcal{D}_{\text{cal}}^{(k)}$. For each repetition $k \in [K]$, after training the machine learning model on $\mathcal{D}_{\text{train}}^{(k)}$ and computing conformity scores on $\mathcal{D}_{\text{cal}}^{(k)}$ and $\mathcal{D}_{\text{test}}$, one can estimate the false discovery proportion corresponding to the rejection of all test points with scores above a fixed rejection threshold $t \in \mathbb{R}$, similarly to (1), with:

$$\widehat{\text{FDP}}^{(k)}(t) = \frac{n_{\text{test}}}{n_{\text{cal}}} \cdot \frac{\sum_{i \in \mathcal{D}_{\text{cal}}^{(k)}} \mathbb{I}\{\hat{S}_i^{(k)} \geq t\}}{\sum_{j \in \mathcal{D}_{\text{test}}} \mathbb{I}\{\hat{S}_j^{(k)} \geq t\}}. \tag{3}$$

Note that the estimate in (3) differs slightly from that in (1) as it lacks the "+1" constant term in the numerator and denominator. While it is possible to include the "+1" terms in (3), this is not needed by our theory and we have observed that it often makes our method unnecessarily conservative. For any fixed $\alpha_{\text{bh}} \in (0,1)$, let $\hat{t}^{(k)}$ be the corresponding BH threshold (2) at the nominal FDR level $\alpha_{\text{bh}}$:

$$\hat{t}^{(k)} = \min\{t \in \tilde{\mathcal{D}}_{\text{cal}-\text{test}}^{(k)} : \widehat{\text{FDP}}^{(k)}(t) \leq \alpha_{\text{bh}}\}, \tag{4}$$

where $\tilde{\mathcal{D}}_{\text{cal}-\text{test}}^{(k)} = \{\hat{S}_i^{(k)}\}_{i \in \mathcal{D}_{\text{test}} \cup \mathcal{D}_{\text{cal}}^{(k)}}$. For each test point $j \in \mathcal{D}_{\text{test}}$, define the following rescaled indicator of whether $\hat{S}_j^{(k)}$ exceeds $\hat{t}^{(k)}$:

$$e_j^{(k)} = (1 + n_{\text{cal}}) \cdot \frac{\mathbb{I}\{\hat{S}_j^{(k)} \geq \hat{t}^{(k)}\}}{1 + \sum_{i \in \mathcal{D}_{\text{cal}}^{(k)}} \mathbb{I}\{\hat{S}_i^{(k)} \geq \hat{t}^{(k)}\}}. \tag{5}$$

Intuitively, this quantifies not only whether the $j$-th null hypothesis would be rejected by the BH filter at the nominal FDR level $\alpha_{\text{bh}}$, but also how extreme $\hat{S}_j^{(k)}$ is relative to the calibration scores. In other words, a large $e_j^{(k)}$ suggests that the test point may be an outlier, where this variable can take any of the following values: 0, 1, $(1 + n_{\text{cal}})/n_{\text{cal}}$, $(1 + n_{\text{cal}})/(n_{\text{cal}} - 1), \ldots, (1 + n_{\text{cal}})$. This approach, inspired by Ren and Barber (2023), is not the only possible way of constructing e-values to derandomize conformal inferences, as discussed in Supplementary Section S5. However, we will show that it works well in practice and it typically achieves higher power compared to standard p-to-e calibrators (Vovk and Wang, 2021) applied to conformal p-values. This advantage partly derives from the fact that (5) can gather strength from many different test points, and partly from the fact that it is not a proper e-value according to the original definition of Vovk and Wang (2021), in the sense that its expected value may be larger than one even if $X_j$ is an inlier. Instead, we will show that our e-values satisfy a relaxed *average validity* property (Ren and Barber, 2023) that is sufficient to guarantee FDR control while allowing more numerous discoveries.

After evaluating (5) for all $j \in \mathcal{D}_{\text{test}}$ and all $k \in [K]$, we aggregate the evidence against the $j$-th null hypothesis into a single statistic $\bar{e}_j$ by taking a weighted average:

$$\bar{e}_j = \sum_{k=1}^{K} w^{(k)} e_j^{(k)}, \qquad \sum_{k=1}^{K} w^{(k)} = 1,$$

based on some appropriate normalized weights $w^{(k)}$. Intuitively, the role of $w^{(k)}$ is to allow for the possibility that the machine learning models based on different realizations of the training subset may not all be equally powerful at separating inliers from outliers. In the remainder of this section, we will take these weights to be known a-priori for all $k \in [K]$, thus representing relevant *side information*; e.g., in the sense of Genovese et al. (2006) and Ren and Candès (2023). For simplicity, one may think for the time being of trivial uninformative weights $w^{(k)} = 1/K$. Of course, it would be preferable to allow these weights to be data-driven, but such an extension is deferred to Section 3.2 for conciseness.

Having calculated aggregate *e-values* $\bar{e}_j$ with the procedure described above, which is outlined by Algorithm S1 in the Supplementary Material, our method rejects the null hypothesis for all $j \in \mathcal{D}_{\text{test}}$ whose $\bar{e}_j$ is greater than an adaptive threshold calculated by applying the eBH filter of Wang and

Ramdas (2022), which is outlined for completeness by Algorithm S2 in the Supplementary Material. We refer to Wang and Ramdas (2022) for a more detailed discussion of the eBH filter. Here, it suffices to recall that the eBH filter computes an adaptive rejection threshold based on the $n_{\text{test}}$ input e-values and on the desired FDR level $\alpha \in (0, 1)$. Then, our following result states that the overall procedure is guaranteed to control the FDR below $\alpha$, under a relatively mild exchangeability assumption.

**Assumption 3.1.** The inliers in $\mathcal{D}$ and the null test points are exchangeable conditional on the non-null test points.

**Theorem 3.2.** *Suppose Assumption 3.1 holds. Then, the e-values computed by Algorithm S1 satisfy:*

$$\sum_{j \in \mathcal{D}_{\text{test}}^{\text{null}}} \mathbb{E}\left[\bar{e}_j\right] \leq n_{\text{test}}. \tag{6}$$

The proof of Theorem 3.2 is in the Supplementary Section S2. Combined with Theorem 2 from Ren and Barber (2023), this result implies our method controls the FDR below the desired target level $\alpha$.

**Corollary 3.3** (Ren and Barber (2023))**.** *The eBH filter of Wang and Ramdas (2022) applied at level $\alpha \in (0, 1)$ to e-values $\{\bar{e}_j\}_{j \in \mathcal{D}_{\text{test}}}$, satisfying (6) guarantees FDR $\leq \alpha$.*

*Remark* 3.4. Assumption 3.1 does not require that the inliers are independent of the outliers.

*Remark* 3.5. Theorem 3.2 holds regardless of the value of the hyper-parameter $\alpha_{\text{bh}}$ of Algorithm S1, which appears in (4). See Section 3.4 for further details about the choice of $\alpha_{\text{bh}}$.

## 3.2 Leveraging data-driven weights

Our method can be extended to leverage adaptive weights based on the data in $\mathcal{D}$ and $\mathcal{D}_{\text{test}}$, as long as each weight $w^{(k)}$ is invariant to permutations of the test point with the corresponding calibration samples in $\mathcal{D}_{\text{cal}}^{(k)}$. In other words, we only require that these weights be written in the form of

$$w^{(k)} = \omega(\tilde{\mathcal{D}}_{\text{cal}-\text{test}}^{(k)}). \tag{7}$$

The function $\omega$ may depend on $\mathcal{D}_{\text{train}}^{(k)}$ but not on $\mathcal{D}_{\text{cal}}^{(k)}$ or $\mathcal{D}_{\text{test}}$. An example of a useful weighting scheme satisfying this property is at the end of this section. The general method is summarized by Algorithm S3 in the Supplementary Material, which extends Algorithm S1. This produces e-values that control the FDR in conjunction with the eBH filter of Wang and Ramdas (2022).

**Theorem 3.6.** *Suppose Assumption 3.1 holds. Then, the e-values computed by Algorithm S3 satisfy* (6), *as long as the adaptive weights obey* (7).

An example of a valid weighting function applied in this paper is the following. Imagine having some prior side information suggesting that the proportion of outliers in $\mathcal{D}_{\text{test}}$ is approximately $\gamma \in (0, 1)$. Then, a natural choice to measure the quality of the $k$-th model is to let $\tilde{w}^{(k)} = |\tilde{v}^{(k)}|$, where $\tilde{v}^{(k)}$ is the standard t-statistic for testing the difference in means between the top $\lceil n_{\text{test}} \cdot \gamma \rceil$ largest values in $\tilde{\mathcal{D}}_{\text{cal}-\text{test}}^{(k)}$ and the remaining ones. See Algorithm S5 in the Supplementary Material for further details. Intuitively, Algorithm S5 tends to assign larger weights to models achieving stronger out-of-sample separation between inliers and outliers. Of course, this approach may not always be optimal but different weighting schemes could be easily accommodated within our framework.

## 3.3 Derandomizing `AdaDetect` with `E-AdaDetect`

The requirement discussed in Section 3.2 that the data-adaptive weights should be invariant to permutations of the calibration and test samples is analogous to the idea utilized by `AdaDetect` (Marandon et al., 2022) to train more powerful machine learning models leveraging also the information contained in the test set; see Section 2.3. This implies that Theorem 3.6 remains valid even if our method is implemented based on $K$ machine learning models each trained by looking also at the unordered union of all data points in $\mathcal{D}_{\text{cal}}^{(k)} \cup \mathcal{D}_{\text{test}}$, for each $k \in [K]$. See Algorithm S4 in the Supplementary Material for a detailed implementation of this extension of our method, which we call `E-AdaDetect`.

## 3.4 Tuning the FDR hyper-parameter

As explained in Section 3.1, our method involves a hyper-parameter $\alpha_{\text{bh}}$ controlling the BH thresholds $\hat{t}^{(k)}$ in (4). Intuitively, higher values of $\alpha_{\text{bh}}$ tend to increase the number of both test and calibration

scores exceeding the rejection threshold at each of the $K$ iterations. Such competing effects make it generally unclear whether increasing $\alpha_{\text{bh}}$ leads to larger e-values in (5) and hence higher power. This trade-off was studied by Ren and Barber (2023) while derandomizing the knockoff filter, and they suggested setting $\alpha_{\text{bh}} < \alpha$. In this paper, we adopt $\alpha_{\text{bh}} = \alpha/10$, which we have observed to work generally well in our context, although even higher power can sometimes be obtained with different values of $\alpha_{\text{bh}}$, especially if the number of outliers in the test set is large. While we leave it to future research to determine whether further improvements are possible, it is worth noting that a straightforward extension of our method, not explicitly implemented in this paper, can be obtained by further averaging e-values obtained with different choices of $\alpha_{\text{bh}}$. Such extension does not affect the validity of (6) due to the linearity of expected values.

## 4 Numerical experiments

### 4.1 Setup and performance metrics

This section compares empirically the performance of `AdaDetect` and our proposed derandomized method described in Section 3.3, namely `E-AdaDetect`. Both procedures are deployed using a binary logistic regression classifier (Marandon et al., 2022) as the base predictive model. The reason why we focus on derandomizing `AdaDetect` instead of traditional split-conformal inferences based on a one-class classifier (Bates et al., 2023) is that we have observed that `AdaDetect` often achieves higher power on the data considered in this paper, which makes it a more competitive benchmark. However, additional experiments reporting on the performance of our derandomization method applied in combination with one-class classifiers can be found in the Supplementary Sections S4.2 and S6.2.

As the objective of this paper is to powerfully detect outliers while mitigating algorithmic randomness, we assess the performance of each method over $M = 100$ independent analyses based on the same fixed data and the same test set. For each repetition $m$ of the novelty detection analysis based on the fixed data, we identify a subset $\mathcal{R}^{(m)} \subseteq \mathcal{D}_{\text{test}}$ of likely outliers (the rejected null hypotheses) and evaluate the average power and false discovery proportion, namely

$$\widehat{\text{Power}} = \frac{1}{M} \sum_{m=1}^{M} \frac{|\mathcal{R}^{(m)} \cap \mathcal{D}_{\text{test}}^{\text{non-null}}|}{|\mathcal{D}_{\text{test}}^{\text{non-null}}|}, \qquad \widehat{\text{FDR}} = \frac{1}{M} \sum_{m=1}^{M} \frac{|\mathcal{R}^{(m)} \cap \mathcal{D}_{\text{test}}^{\text{null}}|}{\max\{|\mathcal{R}^{(m)}|, 1\}}, \qquad (8)$$

where $\mathcal{D}_{\text{test}}^{\text{non-null}} = \mathcal{D}_{\text{test}} \setminus \mathcal{D}_{\text{test}}^{\text{null}}$ indicates the true outliers in the test set. The average false discovery proportion defined in (8) is not the FDR, which is the quantity we can theoretically guarantee to control. In fact, $\text{FDR} = \mathbb{E}[\widehat{\text{FDR}}]$, with expectation taken with respect all randomness in the data. Nonetheless, we will see that this average false discovery proportion is also controlled in practice within all data sets considered in this paper. The advantage of this setup is that it makes it natural to estimate algorithmic variability by observing the consistency of each rejection across independent analyses. In particular, after defining $R_{j,m}$ as the indicator of whether the $j$-th null hypothesis was rejected in the $m$-th analysis, we can evaluate the average variance in the rejection events:

$$\widehat{\text{Variance}} = \frac{1}{n_{\text{test}}} \sum_{j=1}^{n_{\text{test}}} \frac{1}{M-1} \sum_{m=1}^{M} \left( R_{j,m} - \bar{R}_j \right)^2, \qquad (9)$$

where $\bar{R}_j = (1/M) \sum_{m=1}^{M} R_{j,m}$. Intuitively, it would be desirable to maximize power while simultaneously minimizing both the average false discovery proportion and the variability. In practice, however, these metrics often compete with one another; hence why we focus on comparing power and variability for methods designed to control the FDR below the target level $\alpha = 0.1$.

### 4.2 Experiments with synthetic data

Synthetic reference and test data consisting of 100-dimensional vectors $X$ are generated as follows. The reference set contains only inliers, drawn i.i.d. from the standard normal distribution with independent components, $\mathcal{N}(0, I_{100})$. Unless specified otherwise, the test set contains 90% inliers and 10% outliers, independently sampled from $\mathcal{N}(\mu, I_{100})$. The first 5 entries of $\mu$ are equal to a constant parameter, to which we refer as the *signal amplitude*, while the remaining 95 entries are zeros. The size of the reference set is $n = 2000$, with 1000 samples in the training subset and 1000 in the calibration subset. The size of the test set is $n_{\text{test}} = 1000$. Both `E-AdaDetect` and `AdaDetect` are applied based on the same logistic regression classifier with default hyper-parameters.

### 4.2.1 The effect of the signal strength

Figure 2 compares the performance of `E-AdaDetect` (applied with $K = 10$) to that of `AdaDetect`, as a function of the signal amplitude. The results confirm both methods control the FDR but ours is less variable, as expected. The comparison becomes more interesting when looking at power: `AdaDetect` tends to detect more outliers on average if the signal strength is low, but `E-AdaDetect` can also outperform by that metric if the signals are strong. This may be explained as follows. If the signal strength is high, most true discoveries produced by `AdaDetect` are relatively stable across different analyses, while false discoveries may be more aleatory, consistently with the illustration of Figure 1. Such situation is ideal for derandomization, which explains why `E-AdaDetect` is able to simultaneously achieve high power and low false discovery proportion. By contrast, if the signals are weak, the true outlier discoveries produced by `AdaDetect` are relatively scarce and unpredictable, thus behaving not so differently from the false findings. In this case, one could argue that stability becomes even more important to facilitate the interpretation of any findings, and that may justify some loss in average power.

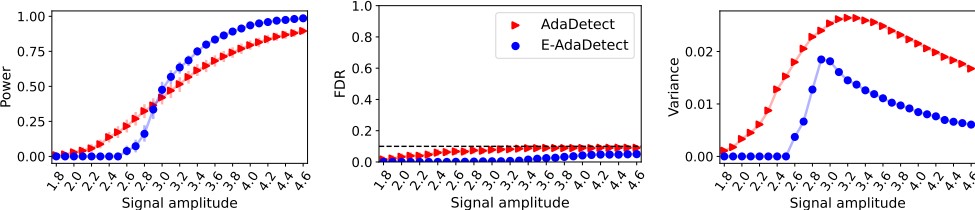

Figure 2: Performance on synthetic data of the proposed derandomized outlier detection method, `E-AdaDetect`, applied with $K = 10$, compared to that of its randomized benchmark, `AdaDetect`, as a function of the signal strength. Both methods leverage a logistic regression binary classifier. Left: average proportion of true outliers that are discovered (higher is better). Center: average proportion of false discoveries (lower is better). Right: variability of the findings (lower is better).

### 4.2.2 The effect of the number of analyses $K$

Figure 3 investigates the effect of varying the number of analyses $K$ aggregated by `E-AdaDetect`. Here, the signal amplitude is fixed to 3.4 (strong signals), while $K$ is varied between 1 and 30. As expected, the results show that the variability of the findings obtained with `E-AdaDetect` decreases as $K$ increases. The average proportion of false discoveries obtained with `E-AdaDetect` also tends to decrease when $K$ is large, which can be understood by noting that spurious findings are less likely to be reproduced consistently across multiple independent analyses of the same data. Regarding power, the average number of true outliers detected by `E-AdaDetect` appears to monotonically increase with $K$, although this is not always true in other situations, as shown in the Supplementary Section S4.1. In fact, if the signals are weak, `E-AdaDetect` may lose some power with larger values of $K$ (although some $K > 1$ may be optimal), consistently with the results shown in Figure 2. Thus, we recommend practitioners to utilize larger values of $K$ in applications where higher power is expected. Finally, note that the power of `E-AdaDetect` is generally lower compared to that of `AdaDetect` in the special case of $K = 1$, although this is not a practically relevant value of $K$ because it does not allow any derandomization. The reason why the power of `E-AdaDetect` is lower when $K = 1$ is that this method relies on the eBH filter. The latter is relatively conservative as an FDR-controlling strategy because it requires no assumptions about the dependencies of the input statistics.

### 4.2.3 The effect of the weighting strategy

This section highlights the practical advantage of being able to use data-adaptive model weights within `E-AdaDetect`. For this purpose, we carry out experiments similar to those of Figure 2, but leveraging a logistic regression model trained with different choices of hyper-parameters in each of the $K$ analyses. Specifically, we fit a sparse logistic regression model using $K = 10$ different values of the regularization parameter. To induce higher variability in the predictive rules, one model was trained with a regularization parameter equal to 0.0001, while the others were trained with regularization parameters equal to 1, 10, 50, and 100, respectively. Then, we apply `E-AdaDetect` using different weighting schemes: constant equal weights ('uniform'), data-driven weights calculated

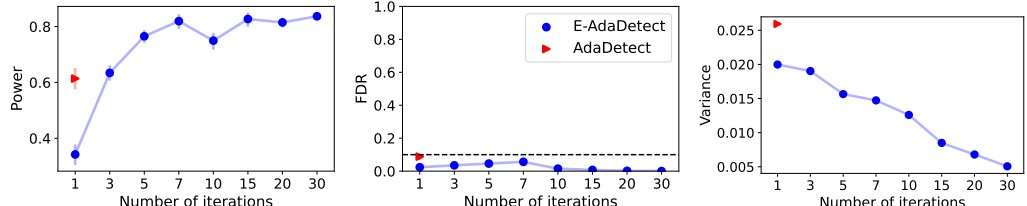

Figure 3: Performance on synthetic data of `E-AdaDetect`, as a function of the number $K$ of derandomized analyses, compared to `AdaDetect`. Note that the latter can only be applied with a single data split (or iteration). The signal amplitude is $3.4$. Other details are as in Figure 2.

with the t-statistic approach ("t-test") summarized by Algorithm S5, and a simple alternative *trimmed average* data-driven approach ("avg. score") outlined by Algorithm S6 in the Supplementary Material. The results in Figure 4 show that the data-driven aggregation scheme based on t-statistics is the most effective one, often leading to much higher power. We have chosen not to compare `E-AdaDetect` to the automatic `AdaDetect` hyper-parameter tuning strategy proposed in Section 4.5 of Marandon et al. (2022) because we found that it does not perform very well in our experiments, possibly due to the relatively low sample size.

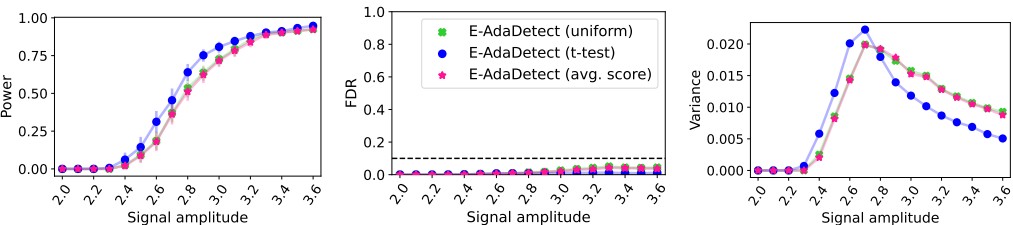

Figure 4: Performance on synthetic data of `E-AdaDetect` applied with different model weighting schemes, as a function of the signal strength. The model weights are designed to prioritize the results of randomized analyses based on models that are more effective at separating inliers from outliers. The t-test approach tends to lead to higher power. Other details are as in Figure 2.

To further demonstrate the effectiveness of data-driven weighting, we turn to analyze the performance of `E-AdaDetect` on four real-world outlier detection data sets: *musk*, *shuttle*, *KDDCup99*, and *credit card*. We refer to Supplementary Section S6 for more information regarding these data. Similar to Figure 4, our `E-AdaDetect` method is applied $K = 10$ times to each data set, each time leveraging a different predictive model as follows. Half of the models are random forests implemented with varying max-depth hyper-parameters (10, 12, 20, 30, and 7), while the other half are support vector machines with an RBF kernel with varying width hyper-parameters (0.1, 0.001, 0.5, 0.2, and 0.03). This setup is interesting because different models often tend to perform differently in practice, and it is usually unclear a-priori which combination of model and hyper-parameters is optimal for a given data set. Figure 5 summarizes the results, demonstrating that both data-driven weighting schemes ("t-test" and "avg. score") lead to more numerous discoveries compared to the "uniform" weighting baseline, and that the "t-test" approach is the most powerful weighting scheme here. These results are in line with the synthetic experiment presented in Figure 4. Lastly, the variance metrics reported in Figure 5 also suggest that data-driven weighting further enhances the algorithmic stability.

### 4.3 Additional results from experiments with synthetic and real data

Sections S4–S6 in the Supplementary Material present the results of several additional experiments. In particular, Section S4 focuses on experiments with synthetic data. Section S5 describes comparisons with alternative derandomization approaches based on different types of p-to-e calibrators (Vovk and Wang, 2021) operating one test point at a time, which turn out to yield lower power compared to our martingale-based method. The results also show that our method compares favorably to an alternative derandomization method based on an e-value construction that first appeared in Ignatiadis et al. (2023), especially if the test set contains numerous outliers or if the nominal FDR level is not

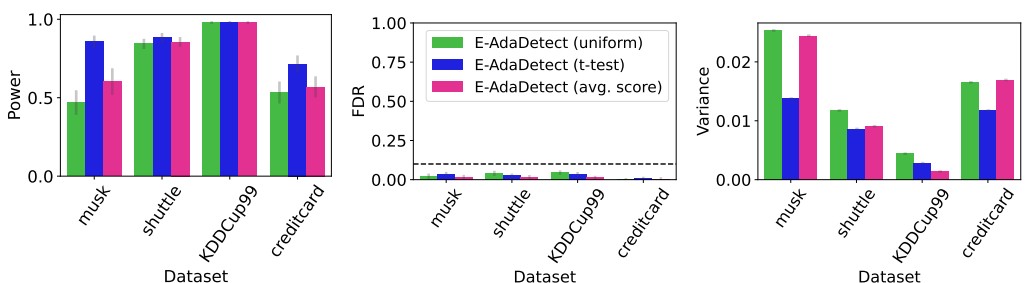

Figure 5: Performance of `E-AdaDetect`, applied with different model weighting schemes, on four real data sets. All methods utilize a combination of random forest and support vector machine models. The "t-test" weighting approach leads to the highest power and lowest algorithmic variability, all while controlling the FDR below the nominal 10% level.

too low. Finally, Section S6 describes additional numerical experiments based on several real data sets also studied in Bates et al. (2023) and Marandon et al. (2022). These results confirm that our martingale-based e-value method can mitigate the algorithmic randomness of standard conformal inferences and `AdaDetect` while retaining relatively high power.

## 5  Discussion

Our experience suggests that e-values are often less powerful than p-values in measuring the statistical evidence against a *single hypothesis*. Yet, e-values can be useful to aggregate multiple dependent tests of the same hypothesis (Vovk and Wang, 2021)—a task that would otherwise require very conservative adjustments within the p-value framework (Vovk et al., 2022). Further, we have shown that e-values lend themselves well to *multiple testing* because they allow efficient FDR control under arbitrary dependence (Wang and Ramdas, 2022), and even their relatively weak *individual* evidence can accumulate rapidly when a large number of hypotheses is probed. The opportunity arising from the combination of these two key properties was recently leveraged to derandomize knockoffs (Ren and Barber, 2023), but until now it had not been fully exploited in the context of conformal inference.

While this paper has focused on derandomizing split-conformal and `AdaDetect` inferences for novelty detection, the key ideas could be easily extended. For example, one may utilize e-values to derandomize conformal prediction intervals in regression (Lei and Wasserman, 2014; Romano et al., 2019) or prediction sets for classification (Lei et al., 2013; Romano et al., 2020) while controlling the false coverage rate over a large test set (Weinstein and Ramdas, 2020). A different direction for future research may explore the derandomization of cross-validation+ (Barber et al., 2021).

We conclude by discussing two limitations of this work. First, the proposed method is more computationally expensive compared to standard conformal inference or `AdaDetect`, and therefore one may be limited to applying it with relatively small numbers $K$ of analysis repetitions when working with very large data sets. That being said, it is increasingly recognized that stability is an important goal in data science (Murdoch et al., 2019), and thus mitigating algorithmic randomness may often justify the deployment of additional computing resources. Second, we have shown that our method sometimes leads to a reduction in the average number of findings compared to randomized alternatives, especially in applications where few discoveries are expected. Therefore, in situations with few anticipated discoveries, practitioners considering applying our method should carefully weigh the anticipated gains in stability versus a possible reduction in power.

Mathematical proofs and additional supporting results are in the Supplementary Material. Software implementing the algorithms described in this paper and enabling the reproduction of the associated numerical experiments is available at https://github.com/Meshiba/derandomized-novelty-detection.

## Acknowledgements

Y. R. and M. B. were supported by the Israel Science Foundation (grant No. 729/21). Y. R. thanks the Career Advancement Fellowship, Technion, for providing research support. Y. R. also thanks Citi Bank for the generous financial support. M. S. was partially supported by NSF grant DMS 2210637 and by an Amazon Research Award.

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
