# Supplementary Material for: "Derandomized novelty detection with FDR control via conformal e-values"

**Meshi Bashari**
Department of Electrical and Computer Engineering
Technion IIT
Haifa, Israel
meshi.b@campus.technion.ac.il

**Amir Epstein**
Citi Innovation Lab
Tel Aviv, Israel
amir.epstein@citi.com

**Yaniv Romano**
Department of Electrical and Computer Engineering
Department of Computer science
Technion IIT
Haifa, Israel
yromano@technion.ac.il

**Matteo Sesia**
Department of Data Sciences and Operations
University of Southern California
Los Angeles, California, USA
sesia@marshall.usc.edu

## Abstract

This document contains mathematical proof, additional details, comparisons to baseline methods, and other supporting information accompanying the paper "Derandomized novelty detection with FDR control via conformal e-values".

## Contents

The supplementary material is organized as follows:

- All algorithmic details are summarized in Section S1.
- Mathematical proofs of theorems presented in the paper can be found in Section S2.
- Section S3 provides details on the training strategy and choice of hyper-parameters for the models utilized in the paper, along with information about the computational resources needed to conduct the experiments.
- Additional synthetic experiments involving our derandomization framework in combination with `AdaDetect` and `OC-Conformal` are in Section S4.
- A discussion of alternative approaches for constructing e-values and corresponding comparisons to our martingale-based e-value construction are in Section S5.
- Real data experiments using `AdaDetect` and `OC-Conformal`, along with their derandomized versions, are in Section S6.

37th Conference on Neural Information Processing Systems (NeurIPS 2023).

# S1 Algorithmic details

---

**Algorithm S1** Aggregation of conformal e-values with fixed model weights

---

1: **Input:** inlier data set $\mathcal{D} \equiv \{X_i\}_{i=1}^n$; test set $\mathcal{D}_{\text{test}}$; size of calibration-set $n_{\text{cal}}$; number of iterations $K$; one-class or binary black-box classification algorithm $\mathcal{A}$; normalized model weights $w^{(k)}$, for $k \in [K]$; hyper-parameter $\alpha_{\text{bh}} \in (0, 1)$;

2: **for** $k = 1, ..., K$ **do**

3:  Randomly split $\mathcal{D}$ into $\mathcal{D}_{\text{cal}}^{(k)}$ and $\mathcal{D}_{\text{train}}^{(k)}$, with $|\mathcal{D}_{\text{cal}}^{(k)}| = n_{\text{cal}}$

4:  Train the model: $\mathcal{M}^{(k)} \leftarrow \mathcal{A}(\mathcal{D}_{\text{train}}^{(k)})$ {possibly including additional labeled outlier data if available}

5:  Compute the calibration scores $S_i^{(k)} = \mathcal{M}^{(k)}(X_i)$, for all $i \in \mathcal{D}_{\text{cal}}^{(k)}$

6:  Compute the test scores $S_j^{(k)} = \mathcal{M}^{(k)}(X_j)$, for all $j \in \mathcal{D}_{\text{test}}$

7:  Compute the threshold $\hat{t}^{(k)}$ according to (4) {this depends on the hyper-parameter $\alpha_{\text{bh}}$ }

8:  Compute the e-values $e_j^{(k)}$ for all $j \in |\mathcal{D}_{\text{test}}|$ according to (5)

9: **end for**

10: Aggregate the e-values $\bar{e}_j = \sum_{k=1}^K w^{(k)} \cdot e_j^{(k)}$

11: **Output:** e-values $\bar{e}_j$ for all $j \in \mathcal{D}_{\text{test}}$ that can be filtered with Algorithm S2 to control the FDR.

---

**Algorithm S2** eBH filter of Wang and Ramdas (2022)

---

1: **Input:** e-values $\{e_j\}_{j=1}^N$ corresponding to $N$ null hypotheses to be tested; target FDR level $\alpha \in (0, 1)$

2: Compute the order statistics of the e-values: $e_{(1)} \geq \cdots \geq e_{(N)}$

3: Find the rejection threshold $i_{\max} = \max\{i \in [N] : e_{(i)} \geq N/(\alpha \cdot i)\}$

4: Construct the rejection set $\mathcal{R} = \{j \in [N] : e_j \geq e_{(i_{\max})}\}$

5: **Output:** a list of rejected null hypotheses $\mathcal{R} \subseteq [N]$.

---

**Algorithm S3** Aggregation of conformal e-values with data-adaptive model weights

---

1: **Input:** inlier data set $\mathcal{D} \equiv \{X_i\}_{i=1}^n$; test set $\mathcal{D}_{\text{test}}$; size of calibration-set $n_{\text{cal}}$; number of iterations $K$; one-class or binary black-box classification algorithm $\mathcal{A}$; a model weighting function $\omega$; hyper-parameter $\alpha_{\text{bh}} \in (0, 1)$;

2: **for** $k = 1, ..., K$ **do**

3:  Randomly split $\mathcal{D}$ into $\mathcal{D}_{\text{cal}}^{(k)}$ and $\mathcal{D}_{\text{train}}^{(k)}$, with $|\mathcal{D}_{\text{cal}}^{(k)}| = n_{\text{cal}}$

4:  Train the model: $\mathcal{M}^{(k)} \leftarrow \mathcal{A}(\mathcal{D}_{\text{train}}^{(k)})$ {possibly including additional labeled outlier data if available}

5:  Compute the calibration scores $S_i^{(k)} = \mathcal{M}^{(k)}(X_i)$, for all $i \in \mathcal{D}_{\text{cal}}^{(k)}$

6:  Compute the test scores $S_j^{(k)} = \mathcal{M}^{(k)}(X_j)$, for all $j \in \mathcal{D}_{\text{test}}$

7:  Compute the weights $\tilde{w}^{(k)} = \omega\left(\{S_i^{(k)}\}_{i \in \mathcal{D}_{\text{test}} \cup \mathcal{D}_{\text{cal}}^{(k)}}\right)$ {invariant un-normalized model weights}

8:  Compute the threshold $\hat{t}^{(k)}$ according to (4) {this depends on the hyper-parameter $\alpha_{\text{bh}}$}

9:  Compute the e-values $e_j^{(k)}$ for all $j \in |\mathcal{D}_{\text{test}}|$ according to (5)

10: **end for**

11: **for** $k = 1, ..., K$ **do**

12:  $w^{(k)} = \tilde{w}^{(k)} / \sum_{k'=1}^K \tilde{w}^{(k')}$ {normalize the model weights}

13: **end for**

14: Aggregate the e-values $\bar{e}_j = \sum_{k=1}^K w^{(k)} \cdot e_j^{(k)}$

15: **Output:** e-values $\bar{e}_j$ for all $j \in \mathcal{D}_{\text{test}}$ that can be filtered with Algorithm S2 to control the FDR.

**Algorithm S4** Aggregation of conformal e-values with data-adaptive model weights and `AdaDetect` training

1: **Input:**  inlier data set $\mathcal{D} \equiv \{X_i\}_{i=1}^n$; test set $\mathcal{D}_{\text{test}}$; size of calibration-set $n_{\text{cal}}$; number of iterations $K$; black-box binary classification algorithm $\mathcal{A}$; a model weighting function $\omega$; hyper-parameter $\alpha_{\text{bh}} \in (0,1)$;
2: **for** $k = 1, ..., K$ **do**
3:     Randomly split $\mathcal{D}$ into $\mathcal{D}_{\text{cal}}^{(k)}$ and $\mathcal{D}_{\text{train}}^{(k)}$, with $|\mathcal{D}_{\text{cal}}^{(k)}| = n_{\text{cal}}$
4:     Train the binary classifier, $\mathcal{M}^{(k)} \leftarrow \mathcal{A}(\mathcal{D}_{\text{train}}^{(k)}, \mathcal{D}_{\text{cal}}^{(k)} \cup \mathcal{D}_{\text{test}})$ {treating the data in $\mathcal{D}_{\text{cal}}^{(k)} \cup \mathcal{D}_{\text{test}}$ as outliers}
5:     Compute the calibration scores $S_i^{(k)} = \mathcal{M}^{(k)}(X_i)$, for all $i \in \mathcal{D}_{\text{cal}}^{(k)}$
6:     Compute the test scores $S_j^{(k)} = \mathcal{M}^{(k)}(X_j)$, for all $j \in \mathcal{D}_{\text{test}}$
7:     Compute the weights $\tilde{w}^{(k)} = \omega\left(\{S_i^{(k)}\}_{i \in \mathcal{D}_{\text{test}} \cup \mathcal{D}_{\text{cal}}^{(k)}}\right)$ {invariant un-normalized model weights}
8:     Compute the threshold $\hat{t}^{(k)}$ according to (4) {this depends on the hyper-parameter $\alpha_{\text{bh}}$}
9:     Compute the e-values $e_j^{(k)}$ for all $j \in |\mathcal{D}_{\text{test}}|$ according to (5)
10: **end for**
11: **for** $k = 1, ..., K$ **do**
12:     $w^{(k)} = \tilde{w}^{(k)} / \sum_{k'=1}^K \tilde{w}^{(k')}$ {normalize the model weights}
13: **end for**
14: Aggregate the e-values $\bar{e}_j = \sum_{k=1}^K w^{(k)} \cdot e_j^{(k)}$
15: **Output:** e-values $\bar{e}_j$ for all $j \in \mathcal{D}_{\text{test}}$ that can be filtered with Algorithm S2 to control the FDR.

---

**Algorithm S5** Adaptive model weighting via t-tests

1: **Input:** Scores $\{S_i\}_{i=1}^N$; a guess $\gamma$ for the proportion of outliers in the data.
2: Compute the order statistics of the scores: $S_{(1)} \leq \cdots \leq S_{(N)}$
3: Denote $n_2 = \lceil \gamma N \rceil$ and $n_1 = N - n_2$
4: Divide the sorted scores into two groups with size $n_1$ and $n_2$: $\mathcal{I}_1 = \{S_{(i)}\}_{i=1}^{n_1}$, $\mathcal{I}_2 = \{S_{(i)}\}_{i=n_1+1}^N$
5: Estimate the means of the two groups: $\mu_1 = (1/n_1) \sum_{i \in \mathcal{I}_1} S_{(i)}$ and $\mu_2 = (1/n_2) \sum_{i \in \mathcal{I}_2} S_{(i)}$
6: Estimate the pooled variance: $z = (1/(n_1 + n_2 - 2)) \left[ \sum_{i \in \mathcal{I}_1} (S_{(i)} - \mu_1)^2 + \sum_{i \in \mathcal{I}_2} (S_{(i)} - \mu_2)^2 \right]$
7: Compute the t-statistic: $\tilde{v} = (\mu_1 - \mu_2)/\sqrt{z \cdot (1/n_1 + 1/n_2)}$
8: **Output:** model weight $\tilde{w} = |\tilde{v}|$

---

**Algorithm S6** Adaptive model weighting via trimmed mean

1: **Input:** Scores $\{S_i\}_{i=1}^N$; a guess $\gamma$ for the proportion of outliers in the data.
2: Compute the order statistics of the scores: $S_{(1)} \leq \cdots \leq S_{(N)}$
3: Denote $\tilde{n} = N - \lceil \gamma N \rceil$
4: Compute the mean of the trimmed group: $\hat{\mu} = \sum_{i=1}^{\tilde{n}} S_{(i)}$
5: **Output:** model weight $\tilde{w} = e^{-\hat{\mu}}$

## S2  Mathematical proofs

*Proof of Theorem 3.2.* This result is implied by Theorem 3.6, to whose proof we refer. $\qquad\square$

*Proof of Theorem 3.6.* The proof follows a martingale argument similar to that of Rava et al. (2021). For each fixed $k$, define the following two quantities as functions of $t \in \mathbb{R}$:

$$V_{\text{test}}^{(k)}(t) = \sum_{j \in \mathcal{D}_{\text{test}}^{\text{null}}} \mathbb{I}\left\{S_j^{(k)} \geq t\right\}, \tag{S1}$$

and

$$V_{\text{cal}}^{(k)}(t) = \sum_{i \in \mathcal{D}_{\text{cal}}^{(k)}} \mathbb{I}\left\{S_i^{(k)} \geq t\right\}. \tag{S2}$$

For each $k \in [K]$, define also the unordered set of conformity scores for non-null test points as:

$$\tilde{\mathcal{D}}_{\text{test}-\text{nn}}^{(k)} = \{\hat{S}_i^{(k)}\}_{i \in \mathcal{D}_{\text{test}} \setminus \mathcal{D}_{\text{test}}^{\text{null}}},$$

and the unordered set of conformity scores for all calibration and test points as:

$$\tilde{\mathcal{D}}_{\text{cal}-\text{test}}^{(k)} = \{\hat{S}_i^{(k)}\}_{i \in \mathcal{D}_{\text{test}} \cup \mathcal{D}_{\text{cal}}^{(k)}}.$$

With this premise, we can write:

$$
\begin{aligned}
\sum_{j \in \mathcal{D}_{\text{test}}^{\text{null}}} \mathbb{E}\left[\bar{e}_j\right] &= \sum_{j \in \mathcal{D}_{\text{test}}^{\text{null}}} \mathbb{E}\left[\sum_{k=1}^{K} w^{(k)} e_j^{(k)}\right] \\
&= \sum_{k=1}^{K} \sum_{j \in \mathcal{D}_{\text{test}}^{\text{null}}} \mathbb{E}\left[w^{(k)} e_j^{(k)}\right] \\
&= \sum_{k=1}^{K} \sum_{j \in \mathcal{D}_{\text{test}}^{\text{null}}} \mathbb{E}\left[w^{(k)} (1 + n_{\text{cal}}) \frac{\mathbb{I}\left\{S_j^{(k)} \geq \hat{t}^{(k)}\right\}}{1 + V_{\text{cal}}^{(k)}(\hat{t}^{(k)})}\right] \\
&= \sum_{k=1}^{K} \mathbb{E}\left[w^{(k)} (1 + n_{\text{cal}}) \mathbb{E}\left[\frac{V_{\text{test}}^{(k)}(\hat{t}^{(k)})}{1 + V_{\text{cal}}^{(k)}(\hat{t}^{(k)})} \mid \tilde{\mathcal{D}}_{\text{cal}-\text{test}}^{(k)}, \tilde{\mathcal{D}}_{\text{test}-\text{nn}}^{(k)}\right]\right] \\
&= \sum_{k=1}^{K} \mathbb{E}\left[w^{(k)} \cdot n_{\text{test}}^{\text{null}}\right] \\
&= n_{\text{test}}^{\text{null}} \leq n_{\text{test}}.
\end{aligned}
$$

Above, the third-to-last equality follows from the assumption that $w^{(k)}$ is a deterministic function of $\tilde{\mathcal{D}}_{\text{cal}-\text{test}}^{(k)}$, and the second-to-last equality follows from the fact that $M^{(k)}(t)$, defined as

$$M^{(k)}(t) = \frac{V_{\text{test}}^{(k)}(t)}{1 + V_{\text{cal}}^{(k)}(t)},$$

is a martingale conditional on $\tilde{\mathcal{D}}_{\text{cal}-\text{test}}^{(k)}$ and $\tilde{\mathcal{D}}_{\text{test}-\text{nn}}^{(k)}$, and therefore it is possible to show that

$$\mathbb{E}\left[M^{(k)}(\hat{t}^{(k)}) \mid \tilde{\mathcal{D}}_{\text{cal}-\text{test}}^{(k)}, \tilde{\mathcal{D}}_{\text{test}-\text{nn}}^{(k)}\right] = \frac{n_{\text{test}}^{\text{null}}}{1 + n_{\text{cal}}}$$

by applying the optional stopping theorem. This last statement is proved below, following the same strategy as in Rava et al. (2021).

For each $l \in \{1, \ldots, n_{\text{test}}^{\text{null}} + n_{\text{cal}}\}$, define $t_l$ as the unique discrete threshold belonging to $\tilde{\mathcal{D}}_{\text{cal}-\text{test}}^{(k)}$ at which exactly $l$ inliers have scores exceeding $t_l$, across all calibration and null test points; i.e.,

$$t_l^{(k)} = \inf\left\{t \in \tilde{\mathcal{D}}_{\text{cal}-\text{test}}^{(k)} : V_{\text{test}}^{(k)}(t) + V_{\text{cal}}^{(k)}(t) = l\right\}.$$

Note that this is always well-defined as long as there are no ties between scores (which can always be achieved by adding a negligible noise).

By convention, define also $t_0^{(k)} = \infty$. Consider then a discrete-time filtration indexed by $l$:

$$\mathcal{F}_l^{(k)} = \left\{ \sigma \left( V_{\text{test}}^{(k)}(t_{l'}^{(k)}), V_{\text{cal}}^{(k)}(t_{l'}^{(k)}) \right) \right\}_{l \leq l' \leq n_{\text{test}}^{\text{null}} + n_{\text{cal}}}.$$

Note that $\mathcal{F}_l^{(k)}$ is a backward-running filtration because $\mathcal{F}_{l_2}^{(k)} \subset \mathcal{F}_{l_1}^{(k)}$ for any $l_1 < l_2$.

It now remains to be proved that $M^{(k)}(t)$ is a martingale. Since we assumed that there are no ties between scores, we get that for every two consecutive thresholds $t_l^{(k)}$ and $t_{l-1}^{(k)}$ the following holds by definition:

$$V_{\text{test}}^{(k)}(t_{l-1}^{(k)}) + V_{\text{cal}}^{(k)}(t_{l-1}^{(k)}) = V_{\text{test}}^{(k)}(t_l^{(k)}) + V_{\text{cal}}^{(k)}(t_l^{(k)}) - 1.$$

The discrepancy between these two thresholds corresponds to a singular score whose value is larger than $t_l^{(k)}$ but smaller than $t_{l-1}^{(k)} > t_l^{(k)}$. This score can either correspond to a calibration or null test point. Therefore, we should consider the following two mutually exclusive events:

$$E_1 = \left\{ V_{\text{cal}}^{(k)}(t_{l-1}^{(k)}) = V_{\text{cal}}^{(k)}(t_l^{(k)}) \right\} \cap \left\{ V_{\text{test}}^{(k)}(t_{l-1}^{(k)}) = V_{\text{test}}^{(k)}(t_l^{(k)}) - 1 \right\},$$

$$E_2 = \left\{ V_{\text{cal}}^{(k)}(t_{l-1}^{(k)}) = V_{\text{cal}}^{(k)}(t_l^{(k)}) - 1 \right\} \cap \left\{ V_{\text{test}}^{(k)}(t_{l-1}^{(k)}) = V_{\text{test}}^{(k)}(t_l^{(k)}) \right\}.$$

By Assumption 3.1,

$$\mathbb{P}\left( E_1 \mid \mathcal{F}_l^{(k)}, \tilde{\mathcal{D}}_{\text{cal}-\text{test}}^{(k)}, \tilde{\mathcal{D}}_{\text{test}-\text{nn}}^{(k)} \right) = \frac{V_{\text{test}}^{(k)}(t_l^{(k)})}{V_{\text{test}}^{(k)}(t_l^{(k)}) + V_{\text{cal}}^{(k)}(t_l^{(k)})},$$

$$\mathbb{P}\left( E_2 \mid \mathcal{F}_l^{(k)}, \tilde{\mathcal{D}}_{\text{cal}-\text{test}}^{(k)}, \tilde{\mathcal{D}}_{\text{test}-\text{nn}}^{(k)} \right) = \frac{V_{\text{cal}}^{(k)}(t_l^{(k)})}{V_{\text{test}}^{(k)}(t_l^{(k)}) + V_{\text{cal}}^{(k)}(t_l^{(k)})}.$$

Then, for any $l \in \{1, \ldots, n_{\text{test}}^{\text{null}} + n_{\text{cal}}\}$, it follows from the law of total probability that

$$\mathbb{E}\left[ M^{(k)}(t_{l-1}^{(k)}) \mid \mathcal{F}_l^{(k)}, \tilde{\mathcal{D}}_{\text{cal}-\text{test}}^{(k)}, \tilde{\mathcal{D}}_{\text{test}-\text{nn}}^{(k)} \right]$$

$$= \frac{V_{\text{test}}^{(k)}(t_l^{(k)}) - 1}{1 + V_{\text{cal}}^{(k)}(t_l^{(k)})} \cdot \frac{V_{\text{test}}^{(k)}(t_l^{(k)})}{V_{\text{cal}}^{(k)}(t_l^{(k)}) + V_{\text{test}}^{(k)}(t_l^{(k)})} + \frac{V_{\text{test}}^{(k)}(t_l^{(k)})}{V_{\text{cal}}^{(k)}(t_l^{(k)})} \cdot \frac{V_{\text{cal}}^{(k)}(t_l^{(k)})}{V_{\text{cal}}^{(k)}(t_l^{(k)}) + V_{\text{test}}^{(k)}(t_l^{(k)})}$$

$$= \frac{V_{\text{test}}^{(k)}(t_l^{(k)}) - 1}{1 + V_{\text{cal}}^{(k)}(t_l^{(k)})} \cdot \frac{V_{\text{test}}^{(k)}(t_l^{(k)})}{V_{\text{cal}}^{(k)}(t_l^{(k)}) + V_{\text{test}}^{(k)}(t_l^{(k)})} + \frac{V_{\text{test}}^{(k)}(t_l^{(k)})}{V_{\text{cal}}^{(k)}(t_l^{(k)}) + V_{\text{test}}^{(k)}(t_l^{(k)})}$$

$$= \frac{V_{\text{test}}^{(k)}(t_l^{(k)}) \cdot [V_{\text{test}}^{(k)}(t_l^{(k)}) - 1] + V_{\text{test}}^{(k)}(t_l^{(k)}) \cdot [1 + V_{\text{cal}}^{(k)}(t_l^{(k)})]}{[1 + V_{\text{cal}}^{(k)}(t_l^{(k)})] \cdot [V_{\text{cal}}^{(k)}(t_l^{(k)}) + V_{\text{test}}^{(k)}(t_l^{(k)})]}$$

$$= \frac{V_{\text{test}}^{(k)}(t_l^{(k)})}{1 + V_{\text{cal}}^{(k)}(t_l^{(k)})}$$

$$= M^{(k)}(t_l^{(k)}).$$

By the optional stopping theorem, this implies that

$$\mathbb{E}\left[ M^{(k)}(\hat{t}^{(k)}) \mid \tilde{\mathcal{D}}_{\text{cal}-\text{test}}^{(k)}, \tilde{\mathcal{D}}_{\text{test}-\text{nn}}^{(k)} \right] = \mathbb{E}\left[ M^{(k)}(t_{n_{\text{test}}^{\text{null}} + n_{\text{cal}}}) \mid \tilde{\mathcal{D}}_{\text{cal}-\text{test}}^{(k)}, \tilde{\mathcal{D}}_{\text{test}-\text{nn}}^{(k)} \right] = \frac{n_{\text{test}}^{\text{null}}}{1 + n_{\text{cal}}},$$

and this completes the proof. $\qquad \square$

## S3 Implementation details for one-class and binary classifiers

We have applied the following models, from the `scikit-learn` (Buitinck et al., 2013) Python library, to compute the scores.

- Synthetic experiments:
  - Binary classifer: logistic regression with `scikit-learn` default ridge regularization parameter.
  - One class classifier: one-class kernel SVM with RBF kernel with `scikit-learn` default kernel width parameter.
- Real-data experiments:
  - Binary classifier: random forest with 100 estimators with a maximum depth of 10. All other hyper-parameters are set to `scikit-learn` default values.
  - One class classifier: isolation forest with 100 estimators, each is fitted to a random subset of $\min(256, n\_\text{samples})$ training samples. All other hyper-parameters are set to `scikit-learn` default values.

Unless specified otherwise, our derandomization method is implemented by setting $\alpha_{\text{bh}} = 0.1 \cdot \alpha$, where $\alpha$ is the target FDR level. All the experiments were conducted on our local CPU cluster.

## S4 Additional synthetic experiments

### S4.1 Derandomized AdaDetect

Section 4.2 of the main manuscript studies the performance of the proposed method focusing on the algorithmic variability for different analyses of the same synthetic data set. Here, we conduct the following additional experiments.

- Figure S1 confirms the reproducibility of the results presented in Figure 2 of the main manuscript, showing that the FDR is controlled over 100 independent realizations of the data. These results are investigated as a function of the signal strength.
- Figure 3 of the main manuscript studies the effect of the number $K$ of derandomized analyses for strong signal amplitude. We conduct a similar study in Figure S2 but for a lower power regime. When the number of iterations is relatively high, our method achieves much smaller algorithmic variably as measured by the selection variance, although at the cost of reduced power. This drop in power can be explained by noting that, in the low power regime, the base outlier detection methods tend to make inconsistent selections across multiple analyses of the same data, which are likely to be filtered out by the derandomization procedure.
- Figure S3 confirms the reproducibility of the results observed in Figure 3 of the main manuscript and Figure S2 of the Supplementary Material, by evaluating the average FDR and power over 100 independent realizations of the data. These results are investigated as a function of the number of analysis repetitions $K$.

### S4.2 Derandomized One-Class Conformal

We now turn to explore the effect of our derandomization method on the performance of `OC-Conformal`. Specifically, Figure S4 compares the power, false discovery proportion, and variance of `OC-Conformal` to those of `E-OC-Conformal` as a function of the signal amplitude, on one realization of the synthetic data. The results show that the false discovery proportion is controlled for both methods, but this error metric is lower for `E-OC-Conformal`. The variance of `E-OC-Conformal` is also reduced, but often at the cost of reduced power. Figure S5 reports the corresponding average FDR and power across 100 realizations, confirming the validity of our method.

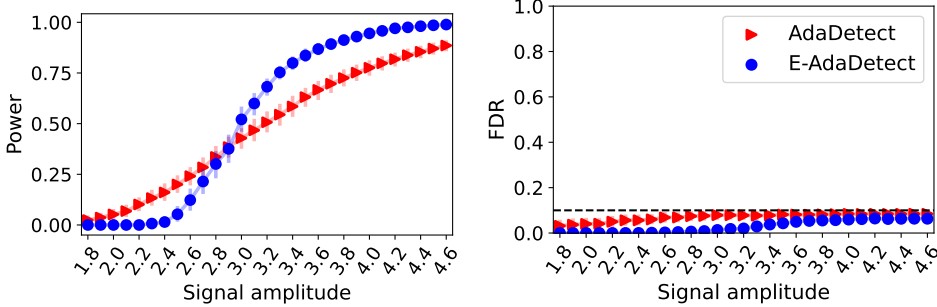

Figure S1: Performance on synthetic data of the proposed derandomized outlier detection method, `E-AdaDetect`, applied with $K = 10$ analysis repetitions. The results are compared to the performance of the randomized benchmark, `AdaDetect`, as a function of the signal strength, averaging over 100 independent realizations of the data. Left: average proportion of true outliers that are discovered (higher is better). Right: average proportion of false discoveries (lower is better). Other results are as in Figure 2.

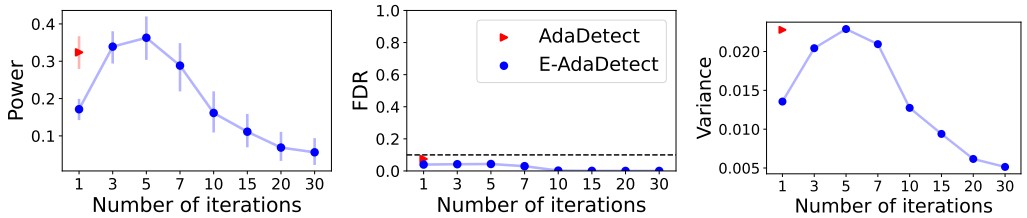

Figure S2: Performance on synthetic data of `E-AdaDetect`, as a function of the number $K$ of analysis repetitions, compared to `AdaDetect`. Note that the latter can only be applied with a single data split (or iteration). Low power regime with signal amplitude 2.8. Other details are as in Figure 3.

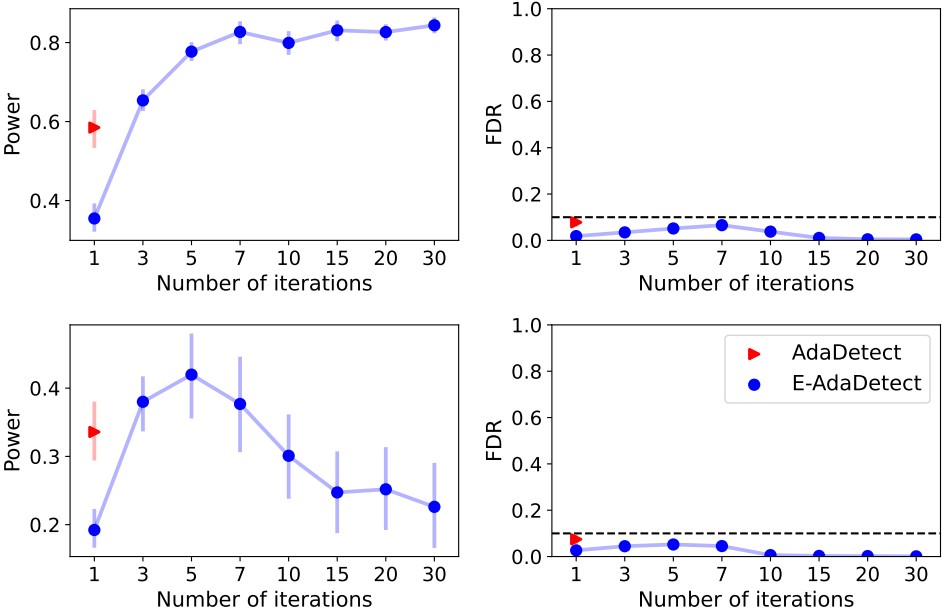

Figure S3: Performance on synthetic data of `E-AdaDetect`, as a function of the number $K$ of analysis repetitions, compared to that of its randomized benchmark `AdaDetect`. The results are averaged over 100 independent realizations of the data. Top: high-power regime with signal amplitude 3.4. Bottom: low-power regime with signal amplitude 2.8. Other results are as in Figure 3.

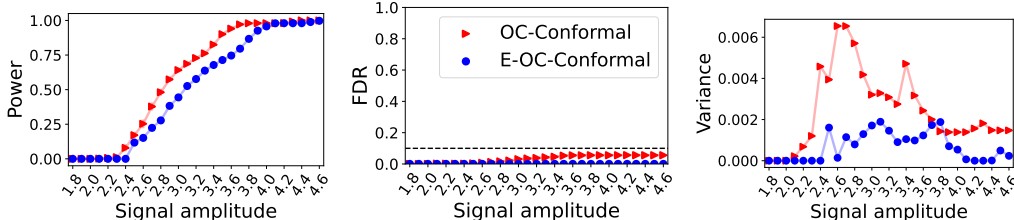

Figure S4: Performance on synthetic data of the proposed derandomized outlier detection method, `E-OC-Conformal`, applied with $K = 10$ analysis repetitions. The results are compared to the performance of the randomized benchmark, `OC-Conformal`, as a function of the signal strength. Both methods leverage a one-class support vector classifier. Left: average proportion of true outliers that are discovered (higher is better). Center: average proportion of false discoveries (lower is better). Right: variability of the findings (lower is better). Other details are as in Figure 2. Note that these results correspond to 100 repeated experiments based on a single realization of the labeled and test data, hence why the results appear a little noisy.

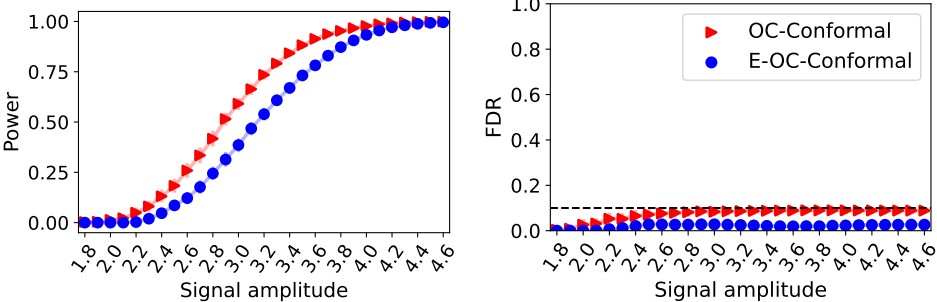

Figure S5: Performance on synthetic data of the proposed derandomized outlier detection method, `E-OC-Conformal`, applied with $K = 10$ analysis repetitions. The results are compared to the performance of the randomized benchmark, `OC-Conformal`, as a function of the signal strength. Both methods leverage a one-class support vector classifier. The results are averaged over 100 independent realizations of the data. Left: average proportion of true outliers that are discovered (higher is better). Right: average proportion of false discoveries (lower is better). Other details are as in Figure 2.

## S4.3  Hyper-parameter tuning

In all of the experiments presented in the main manuscript, we set $\alpha_{\mathrm{bh}}$ to be $\alpha/10$. We found this to be a reasonable choice in general, although it may not always be optimal. In this section, we explore the effect of $\alpha_{\mathrm{bh}}$ across four scenarios: low and high power regimes, as well as small and large proportions of outliers in the test set. Following Figure S7, we can see that the choice $\alpha_{\mathrm{bh}} = \alpha/10$ is not always ideal for the E-OC-Conformal algorithm. Here, a larger value of $\alpha_{\mathrm{bh}} = \alpha/2$ seems to be a better choice when the proportion of outliers is large, as it results in more test outlier samples with non-zero e-values. By contrast, a smaller value of $\alpha_{\mathrm{bh}} = \alpha/10$ is suitable when the proportion of outliers is small, as it leads to larger e-values for the outliers.

We now repeat the same experiment but with E-AdaDetect, summarizing the results in Figure S6. In contrast with E-OC-Conformal, a fixed $\alpha/10$ is an appropriate choice for $\alpha_{\mathrm{bh}}$ when applying our method with AdaDetect for all the scenarios we studied, indicating that this method is more robust to the choice of $\alpha_{\mathrm{bh}}$. In the remaining supplementary experiments, we will utilize a fixed $\alpha_{\mathrm{bh}} = \alpha/2$ for E-OC-Conformal and a fixed $\alpha_{\mathrm{bh}} = \alpha/10$ for E-AdaDetect.

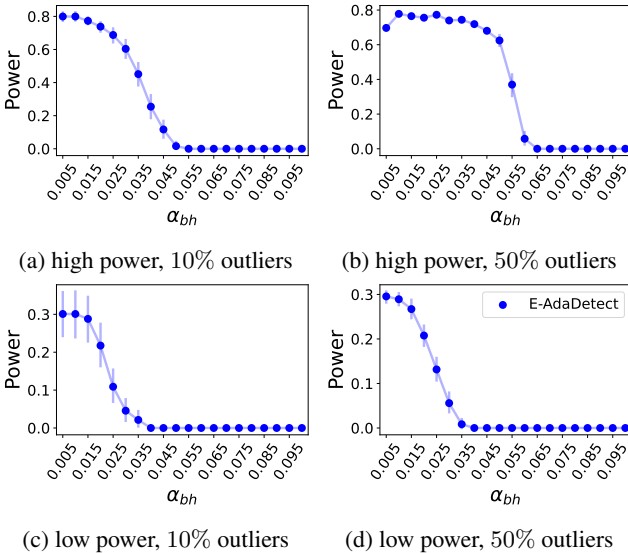

(a) high power, 10% outliers      (b) high power, 50% outliers

(c) low power, 10% outliers      (d) low power, 50% outliers

Figure S6: Performance on synthetic data of the proposed derandomized outlier detection method, E-AdaDetect, applied with $K = 10$ as a function of $\alpha_{\mathrm{bh}}$. The results are averaged over 100 independent realizations of the data. Top: high-power regime with signal amplitude 3.4 for 10% outliers and 1.6 for 50% outliers. Bottom: low-power regime with signal amplitude 2.8 for 10% outliers and 1.1 for 50% outliers. Left: 10% outliers in the test-set. Right: 50% outliers in the test-set. Other details are as in Figure 2.

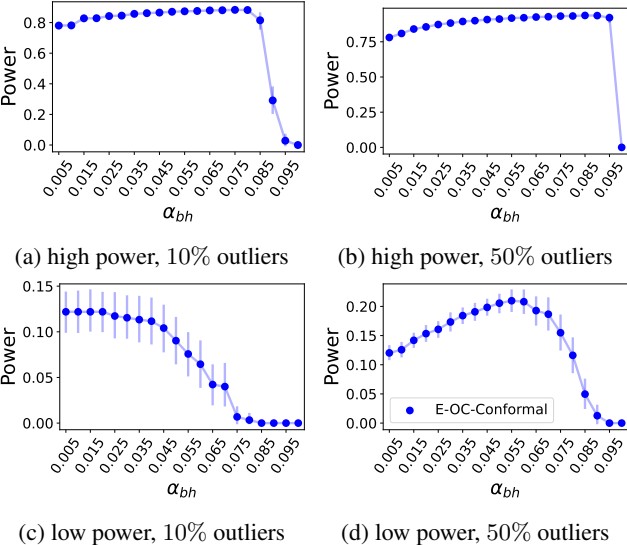

(a) high power, 10% outliers

(b) high power, 50% outliers

(c) low power, 10% outliers

(d) low power, 50% outliers

Figure S7: Performance on synthetic data of the proposed derandomized outlier detection method, `E-OC-Confromal`, applied with $K = 10$ as a function of $\alpha_{\mathrm{bh}}$. The method leverages a one-class support vector classifier. The results are averaged over 100 independent realizations of the data. Top: high-power regime with signal amplitude 3.6 for 10% outliers and 3.4 for 50% outliers. Bottom: low-power regime with signal amplitude 2.6 for 10% outliers and 2.3 for 50% outliers. Left: 10% outliers in the test-set. Right: 50% outliers in the test-set. Other details are as in Figure 2.

## S5 Comparisons to alternative e-values constructions

In this section, we discuss alternative methods for constructing conformal e-values to derandomize split conformal inferences, comparing their performance to that of our proposed martingale-based approach. These alternative e-value constructions were proposed in prior works, as detailed below, but they had not been previously utilized for the purpose of de-randomizing conformal inferences.

### S5.1 Review of p-to-e calibrators

One strategy for generating e-values is to utilize *p-to-e calibrators*, whose goal is to transform valid p-values into valid e-values (Vovk and Wang, 2021, Section 2). This strategy can be specifically applied to conformal p-values $\hat{u}$. Various types of calibrators are available, including Shafer's calibrator:

$$S(\hat{u}) := \frac{1}{\sqrt{\hat{u}}} - 1, \tag{S3}$$

as well as the following family of calibrators,

$$F(\hat{u}) = \epsilon \cdot \hat{u}^{\epsilon-1}, \tag{S4}$$

where $\epsilon \in (0, 1)$ is a hyper-parameter. To bypass the problem of choosing the hyper-parameter $\epsilon$ in (S4), one can use an over-optimistic estimate of the maximum of (S4), known as the VS calibrator (Vovk and Wang, 2021):

$$\text{VS}(\hat{u}) := \sup_{\epsilon} \epsilon \hat{u}^{\epsilon-1} = \begin{cases} -e^{-1}/(\hat{u} \ln \hat{u}), & \text{if } \hat{u} \leq e^{-1} \\ 1, & \text{otherwise} \end{cases}. \tag{S5}$$

The VS calibrator does not produce a valid e-value, but it can still serve as an informative baseline because it approximates the most powerful possible calibrator within the family of functions (S4) (Vovk and Wang, 2021). An alternative way to eliminate the influence of $\epsilon$ is to integrate over it, leading to the following calibrator:

$$F(\hat{u}) := \int_0^1 \epsilon \hat{u}^{\epsilon-1} d\epsilon = \frac{1 - \hat{u} + \hat{u} \ln \hat{u}}{\hat{u}(-\ln \hat{u})^2}. \tag{S6}$$

Armed with a p-to-e calibrator, one can then derandomize split conformal inferences by proceeding similarly to the main manuscript; this approach is summarized for completeness in Algorithm S7.

**Algorithm S7** Aggregation of conformal e-values computed by a p-to-e calibrator with data-adaptive model weights

---

1: **Input:** inlier data set $\mathcal{D} \equiv \{X_i\}_{i=1}^n$; test set $\mathcal{D}_{\text{test}}$; size of calibration-set $n_{\text{cal}}$; number of iterations $K$; p-to-e calibrator function $F$; one-class or binary black-box classification algorithm $\mathcal{A}$; a model weighting function $\omega$;

2: **for** $k = 1, ..., K$ **do**

3:      Randomly split $\mathcal{D}$ into $\mathcal{D}_{\text{cal}}^{(k)}$ and $\mathcal{D}_{\text{train}}^{(k)}$, with $|\mathcal{D}_{\text{cal}}^{(k)}| = n_{\text{cal}}$

4:      Train the model: $\mathcal{M}^{(k)} \leftarrow \mathcal{A}(\mathcal{D}_{\text{train}}^{(k)})$ {possibly including additional labeled outlier data if available}

5:      Compute the calibration scores $S_i^{(k)} = \mathcal{M}^{(k)}(X_i)$, for all $i \in \mathcal{D}_{\text{cal}}^{(k)}$

6:      Compute the test scores $S_j^{(k)} = \mathcal{M}^{(k)}(X_j)$, for all $j \in \mathcal{D}_{\text{test}}$

7:      Compute the weights $\tilde{w}^{(k)} = \omega\left(\{S_i^{(k)}\}_{i \in \mathcal{D}_{\text{test}} \cup \mathcal{D}_{\text{cal}}^{(k)}}\right)$ {invariant un-normalized model weights}

8:      Compute the p-values for all $j \in |\mathcal{D}_{\text{test}}| : \hat{u}_j^{(k)} = (1 + \sum_{i \in \mathcal{D}_{\text{cal}}} \mathbb{I}\{S_j^{(k)} \leq S_i^{(k)}\})/(1 + n_{\text{cal}})$

9:      Compute the e-values $e_j^{(k)}$ for all $j \in |\mathcal{D}_{\text{test}}|$ using the p-to-e calibrator: $e_j^{(k)} = F\left(\hat{u}_j^{(k)}\right)$

10: **end for**

11: **for** $k = 1, ..., K$ **do**

12:      $w^{(k)} = \tilde{w}^{(k)} / \sum_{k'=1}^K \tilde{w}^{(k')}$ {normalize the model weights}

13: **end for**

14: Aggregate the e-values $\bar{e}_j = \sum_{k=1}^K w^{(k)} \cdot e_j^{(k)}$

15: **Output:** e-values $\bar{e}_j$ for all $j \in \mathcal{D}_{\text{test}}$ that can be filtered with Algorithm S2 to control the FDR.

---

### S5.1.1    Comparing p-to-e to the martingale-based approach

When implementing the p-to-e derandomization approach in the synthetic experiments described in Section 4.2, we observed that the power was nearly zero. Increasing the size of the calibration set can be beneficial to improve the power of this approach. This is because the size of the calibration set determines the minimum attainable conformal p-value, given by $1/(n_{\text{cal}} + 1)$. Consequently, the size of this set influences the maximum achievable e-value through p-to-e calibrators: smaller input p-values result in larger outputs from the calibrator functions (S3), (S5), and (S6).

Following the above discussion, we compare the performance of the p-to-e approach to our martingale-based method as a function of the size of the calibration set. According to Figure S8, we can see that the p-to-e approach has lower power than our method, where both derandomization methods are combined with `OC-Conformal`. Among the studied p-to-e calibrators, the VS calibrator demonstrates relatively higher power. However, it should be noted that the VS calibrator generates invalid e-values, as this calibrator outputs an overly optimistic estimate of the maximum value in (S4). Nevertheless, even the VS calibrator is considerably less powerful than our proposed method.

Since there is a significant performance gap between our martingale-based e-value construction and the p-to-e approaches, we do not provide any further comparisons between these methods. We also do not repeat this experiment with `AdaDetect`, since the p-to-e approach requires a large calibration set to yield meaningful power, which is far from an ideal setup for `AdaDetect`. The latter approach suggests fitting a binary classifier on the observed data, treating both the calibration and test points as outliers. A large amount of inlier calibration points can lower `AdaDetect`'s power since the wrong labeling of the calibration points as outliers is likely to reduce the classifier's ability to provide large scores for test outliers.

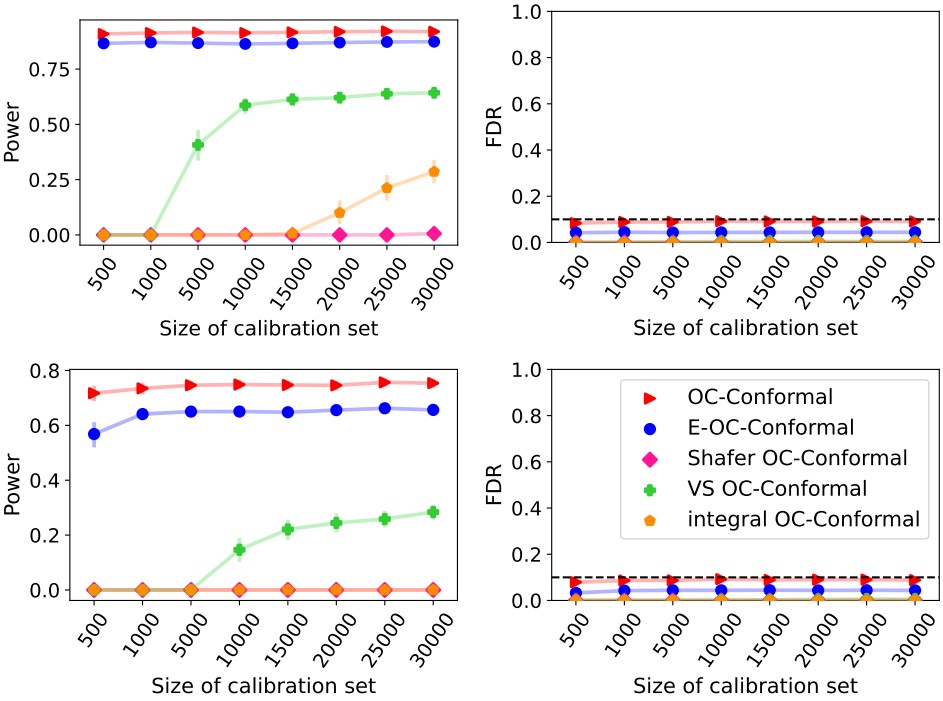

Figure S8: Performance on synthetic data of the proposed derandomized outlier detection method, `E-OC-Conformal`, applied with $K = 10$, compared to that of its randomized benchmark, `OC-Conformal`. We also compare the performance of these methods to p-to-e calibrators, applied with $K = 10$, as a function of the number of inlier calibration points. The number of training inliers is fixed and equals 1000. All methods leverage a one-class support vector classifier. Top: high-power regime with signal amplitude 3.6. Bottom: low-power regime with signal amplitude 3.2. The dashed horizontal line indicates the nominal false discovery rate level $\alpha = 0.1$. The results are averaged over 100 independent realizations of the data.

## S5.2    Review of soft-rank permutation e-test

The soft-rank e-values introduced by Ignatiadis et al. (2023) is another approach to construct e-values for permutation tests, including split conformal. This method constructs an e-value for each test point by comparing its relative rank to the calibration samples, employing a similar methodology to the construction of conformal p-values described in Bates et al. (2023). Here, we present a slightly modified version that begins with normalizing the conformity score for each test point as well as the calibration scores. Consider a single hypothesis corresponding to a single test point. Let $S_0$ be the corresponding test conformity score, and $S_1, \ldots, S_{n_{\text{cal}}}$ be the $n_{\text{cal}}$ conformity scores correspond to the calibration set. With this in place, denote by $S_{\text{max}}$ and $S_{\text{min}}$ the maximum and minimum scores among $S_0, \ldots, S_{n_{\text{cal}}}$. Then, for each $b \in [0, n_{\text{cal}}]$ we define the normalized score as

$$L_b = \frac{S_b - S_{\text{min}}}{S_{\text{max}} - S_{\text{min}}}. \tag{S7}$$

Having defined the normalized score, we construct an e-value for each test point by following the set of steps described in Ignatiadis et al. (2023). Define $L_* = \min_{b=0,\ldots,n_{\text{cal}}} L_b$. For $b = 0, \ldots, n_{\text{cal}}$, compute the transformed statistic as

$$R_b = \frac{e^{rL_b} - e^{rL_*}}{r}, \tag{S8}$$

where $r > 0$ is a hyper-parameter. In the case where $r = 0$, the transformed statistic simplifies to $R_b = L_b - L_*$. Overall, the soft-ranking transformation presented above preserves the ordering of the test statistics while ensuring that the random variable $R_b$ is non-negative. Leveraging the transformed non-negative variables, we can construct a valid e-value for the test point by computing (Ignatiadis

et al., 2023)

$$e_0 := (n_{\text{cal}} + 1) \frac{R_0}{\sum_{i=0}^{n_{\text{cal}}} R_i}. \tag{S9}$$

Similarly to p-to-e calibrators, we can combine the soft-rank e-value approach with our novel derandomization procedure, as outlined in Algorithm S8, and further compare the performance of this method to our martingale-based e-values.

Before doing so, we pause to discuss the choice of the hyper-parameter $r$. Since there is no simple rule on how to set this parameter, we repeat the same analysis from Section S4.3 and study the effect of $r$ across four scenarios: low/high power regimes and small/large proportions of test outliers. The results, presented in Figures S9 and S10, suggest that a suitable choice for $r$ could be $500$ for AdaDetect and $r = 75$ for OC-Conformal. The latter choice takes into account the trade-off in power across Figures S10b and S10c. We use these choices for all the soft-rank experiments provided in this Supplementary Material.

---

**Algorithm S8** Aggregation of soft-rank e-values with data-adaptive model weights

---

1: **Input:** inlier data set $\mathcal{D} \equiv \{X_i\}_{i=1}^n$; test set $\mathcal{D}_{\text{test}}$; size of calibration-set $n_{\text{cal}}$; number of iterations $K$; one-class or binary black-box classification algorithm $\mathcal{A}$; a model weighting function $\omega$; hyper-parameter $r \in [0, \infty)$;

2: **for** $k = 1, ..., K$ **do**

3:     Randomly split $\mathcal{D}$ into $\mathcal{D}_{\text{cal}}^{(k)}$ and $\mathcal{D}_{\text{train}}^{(k)}$, with $|\mathcal{D}_{\text{cal}}^{(k)}| = n_{\text{cal}}$

4:     Train the model: $\mathcal{M}^{(k)} \leftarrow \mathcal{A}(\mathcal{D}_{\text{train}}^{(k)})$ {possibly including additional labeled outlier data if available}

5:     Compute the calibration scores $S_i^{(k)} = \mathcal{M}^{(k)}(X_i)$, for all $i \in \mathcal{D}_{\text{cal}}^{(k)}$

6:     Compute the test scores $S_j^{(k)} = \mathcal{M}^{(k)}(X_j)$, for all $j \in \mathcal{D}_{\text{test}}$

7:     Compute the weights $\tilde{w}^{(k)} = \omega\left(\{S_i^{(k)}\}_{i \in \mathcal{D}_{\text{test}} \cup \mathcal{D}_{\text{cal}}^{(k)}}\right)$ {invariant un-normalized model weights}

8:     Normalize the scores according to (S7)

9:     Compute the transformed score for all $j \in |\mathcal{D}_{\text{test}}|$ according to (S8) {this depends on the hyper-parameter $r$}

10:     Compute the e-values $e_j^{(k)}$ for all $j \in |\mathcal{D}_{\text{test}}|$ according to (S9)

11: **end for**

12: **for** $k = 1, ..., K$ **do**

13:     $w^{(k)} = \tilde{w}^{(k)} / \sum_{k'=1}^K \tilde{w}^{(k')}$ {normalize the model weights}

14: **end for**

15: Aggregate the e-values $\bar{e}_j = \sum_{k=1}^K w^{(k)} \cdot e_j^{(k)}$

16: **Output:** e-values $\bar{e}_j$ for all $j \in \mathcal{D}_{\text{test}}$ that can be filtered with Algorithm S2 to control the FDR.

---

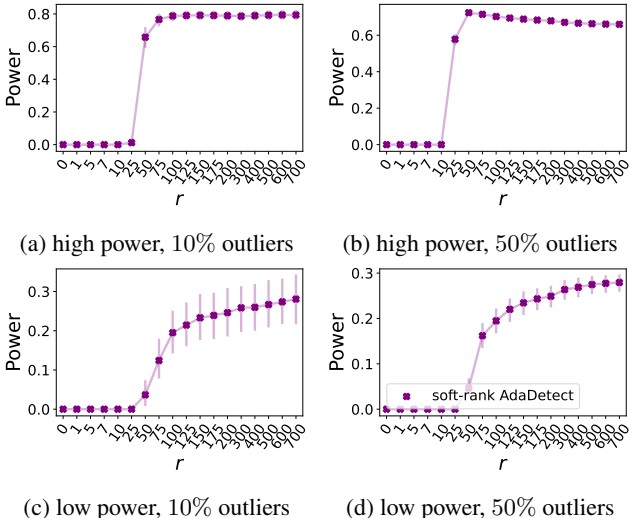

(a) high power, 10% outliers  (b) high power, 50% outliers

(c) low power, 10% outliers  (d) low power, 50% outliers

Figure S9: Performance on synthetic data of the proposed derandomized outlier detection method applied with soft-rank e-values, `soft-rank OC-Conformal`, applied with $K = 10$ as a function of $r$ hyper-parameter. The results are averaged over 100 independent realizations of the data. Top: high-power regime with signal amplitude 3.4 for 10% outliers and 1.6 for 50% outliers. Bottom: low-power regime with signal amplitude 2.8 for 10% outliers and 1.1 for 50% outliers. Left: 10% outliers in the test-set. Right: 50% outliers in the test-set. Other details are as in Figure 2.

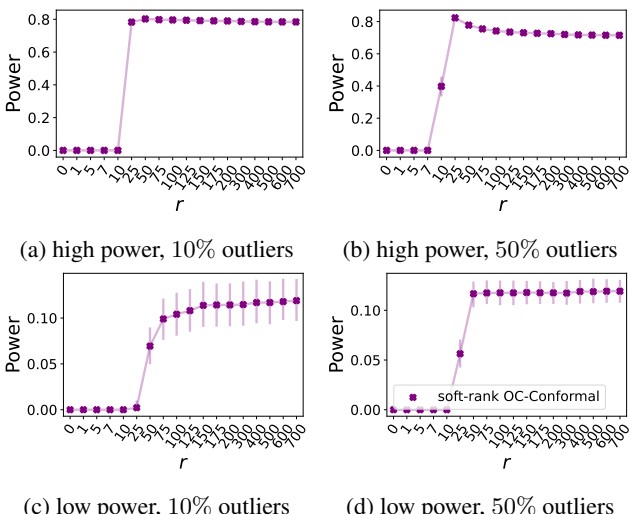

(a) high power, 10% outliers  (b) high power, 50% outliers

(c) low power, 10% outliers  (d) low power, 50% outliers

Figure S10: Performance on synthetic data of the proposed derandomized outlier detection method applied with soft-rank e-values, `soft-rank OC-Conformal`, applied with $K = 10$ as a function of $r$ hyper-parameter. The method leverages a one-class support vector classifier. The results are averaged over 100 independent realizations of the data. Top: high-power regime with signal amplitude 3.6 for 10% outliers and 3.4 for 50% outliers. Bottom: low-power regime with signal amplitude 2.6 for 10% outliers and 2.3 for 50% outliers. Left: 10% outliers in the test set. Right: 50% outliers in the test set. Other details are as in Figure 2.

### S5.2.1  Comparing soft-rank e-values to the martingale-based e-values

In striking contrast with the soft-rank e-values that are constructed separately for each test point, our martingale-based e-values are constructed jointly by looking at all test scores. Intuitively, by leveraging the additional information present in the test set, the martingale-based e-values may achieve higher power. Mathematically, recall that the soft-rank e-value is valid by construction,

implying that $\mathbb{E}[e] \leq 1$ under the null hypothesis. By contrast, our martingale-based e-values satisfy a more relaxed property for which $\sum_{j \in \mathcal{D}_{\text{test}}^{\text{null}}} \mathbb{E}[e_j] \leq n_{\text{test}}$. Consequently, in settings where the proportion of test outliers is large, each of the inlier e-values can exceed the value 1 as long as their sum is bounded by $n_{\text{test}}$, in expectation. This can be attractive since we anticipate the non-null e-values that correspond to outlier points to have larger values than the null ones. Indeed, the following experiments indicate that the martingale-based approach tends to be more powerful than the soft-rank e-values when the proportion of outliers in the test set is relatively large.

In more detail, we compare in Figure S11 the soft-rank approach with our martingale-based method by varying the proportion of outliers present in the test set. That figure is obtained by adjusting the signal amplitude level such that the power of the randomized method (`AdaDetect/OC-Conformal`) is fixed at around 80% for all the range of outlier proportions we studied. It is evident from that figure that the gap between the soft-rank e-value and our martingale-based e-value increases as the proportion of outliers increases, and that our proposal is more powerful than the soft-rank approach. For completeness, a comprehensive comparison considering various proportions of outliers as a function of the signal amplitude can be found in Figure S12 and Figure S13.

We also investigate the impact of the target FDR level on the performance of the soft-rank and our martingale-based methods. The performance metrics, shown in Figures S14 and S15, reveal that our method is more powerful, displaying greater adaptability to the FDR level. This aligns with our expectations, as our hyper-parameter $\alpha_{\text{bh}}$ is set proportionally to the target FDR level $\alpha$. By contrast, the soft-rank hyper-parameter $r$ remains fixed across different target FDR levels; it is unclear how to refine the choice of this parameter in this setting, or even to conclude whether the best choice of $r$ is affected by the target FDR level.

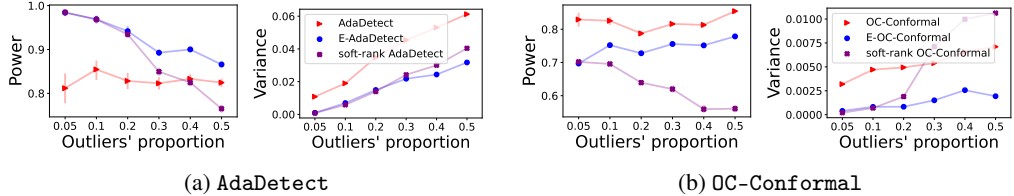

(a) `AdaDetect`

(b) `OC-Conformal`

Figure S11: Performance on synthetic data of the proposed derandomized outlier detection method, `E-AdaDetect (E-OC-Conformal)`, applied with $K = 10$, compared to that of its randomized benchmark, `AdaDetect (OC-Conformal)`. We also compare these methods to the soft-rank method, applied with $K = 10$, `soft-rank AdaDetect (soft-rank OC-Conformal)`, as a function of the proportion of outliers in the test set with the corresponding signal strength that results in a stable strength of the randomized benchmarks.

## S6    Experiments with real data

### S6.1    Derandomized AdaDetect

In this section, we evaluate the performance of our method on several benchmark data sets for outlier detection, also studied in Bates et al. (2023) and Marandon et al. (2022): *musk* (mus), *shuttle* (shu), *KDDCup99* (KDD), and *credit card* (cre). We refer to Bates et al. (2023) and Marandon et al. (2022) for more details about these data sets. Similarly to Section 4.2, we construct a reference set and a test set through random sub-sampling. The reference set contains 3000 inliers, and the test set contains 1000 samples, of which we control the proportion of outliers. We apply our derandomization procedure using the proposed martingale-based e-values and soft-rank e-values implemented in combination with `AdaDetect`, using $K = 10$ independent splits of the reference set into training and calibration subsets of size 2000 and 1000, respectively. All the methods are repeatedly applied to carry out 100 independent analyses of the same data. In the regime that the test set contains 10% outliers, we can see from Figure S16a that all methods control the average proportion of false discoveries below $\alpha = 0.1$ and achieve similar power, but the findings obtained with the derandomized methods are far more stable. By contrast, when increasing the proportion of outliers to $40\%$ (Figure S16b) we can see that the martingale-based approach tends to be more powerful than the soft-rank method. Finally, Figure S18 confirms the reproducibility of these results by reporting the average FDR and

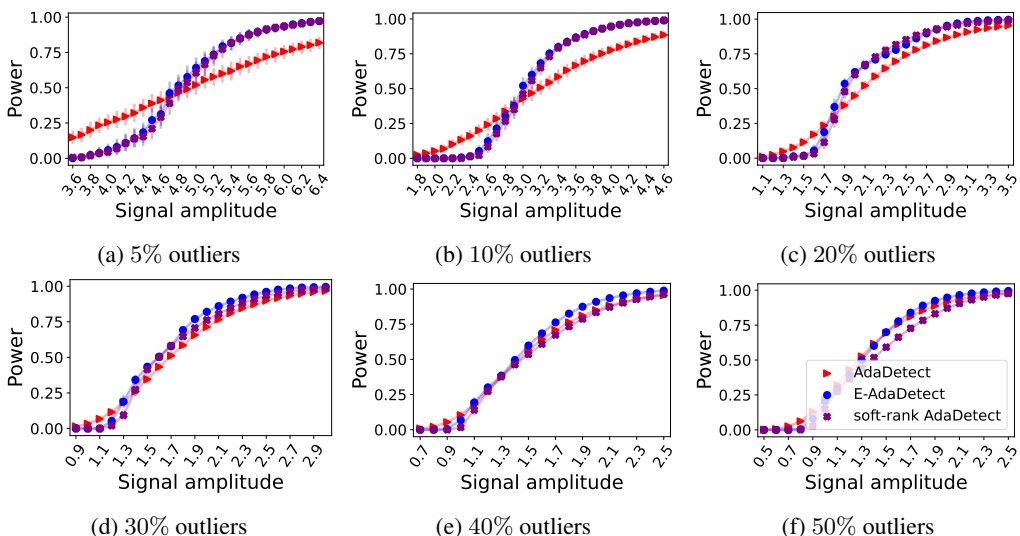

(a) 5% outliers     (b) 10% outliers     (c) 20% outliers

(d) 30% outliers     (e) 40% outliers     (f) 50% outliers

Figure S12: Performance on synthetic data of the proposed derandomized outlier detection method, `E-AdaDetect`, applied with $K = 10$, compared to that of its randomized benchmark, `AdaDetect`. We compare these methods to the soft-rank method, applied with $K = 10$, `soft-rank AdaDetect`, as a function of the signal strength for varying proportions of outliers in the test set. The results are averaged over 100 independent realizations of the data. Other results are as in Figure 2.

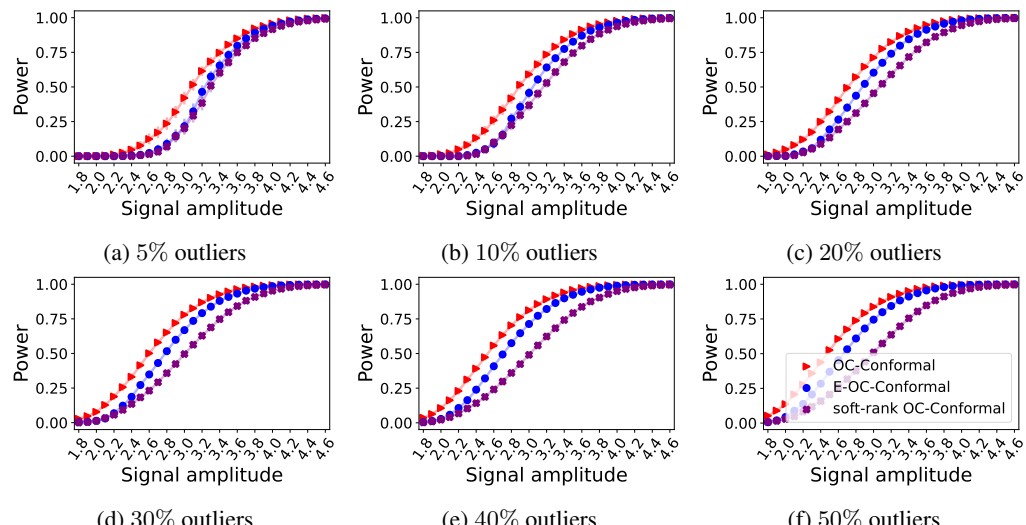

(a) 5% outliers     (b) 10% outliers     (c) 20% outliers

(d) 30% outliers     (e) 40% outliers     (f) 50% outliers

Figure S13: Performance on synthetic data of the proposed derandomized outlier detection method, `E-OC-Conformal`, applied with $K = 10$, compared to that of its randomized benchmark, `OC-Conformal`. These methods are also compared to the soft-rank method, applied with $K = 10$, `soft-rank OC-Conformal`, as a function of the signal strength for varying proportion of outliers in the test set. The results are averaged over 100 independent realizations of the data. Other results are as in Figure S5.

power over 100 independent realizations of the sub-sampled data considered in Figure S16; these performance metrics are presented as a function of the outlier proportion.

## S6.2   Derandomized One-Class Conformal

We turn to study the effect of our approach on `OC-Conformal` on the same real data sets, by following the experimental protocol from Section S6.1. In general, we observe that `OC-Conformal` is less

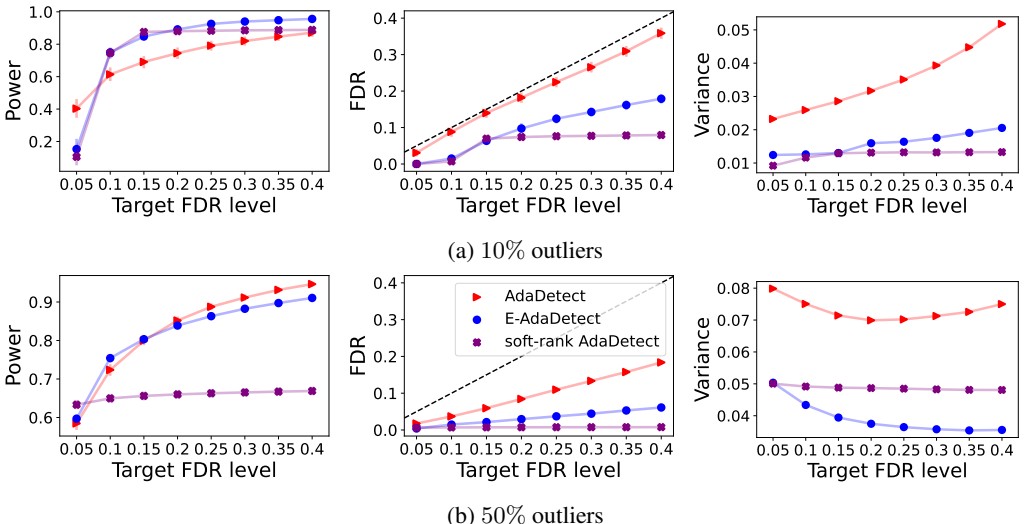

(a) 10% outliers

(b) 50% outliers

Figure S14: Performance on synthetic data of the proposed derandomized outlier detection method, `E-AdaDetect`, applied with $K = 10$, compared to that of its randomized benchmark, `AdaDetect`. These methods are also compared to the soft-rank method, applied with $K = 10$, `soft-rank AdaDetect`, as a function of the target FDR level. All methods leverage a logistic regression binary classifier. S14a presents the performance in high-power regime with signal amplitude $3.4$ when there are 10% outliers. S14b presents the performance in high-power regime with signal amplitude $1.6$ when there are 50% outliers. The dashed line indicates the nominal false discovery rate level. Note that these results correspond to 100 repeated experiments based on a single realization of the labeled and test data, hence why the results appear a little noisy.

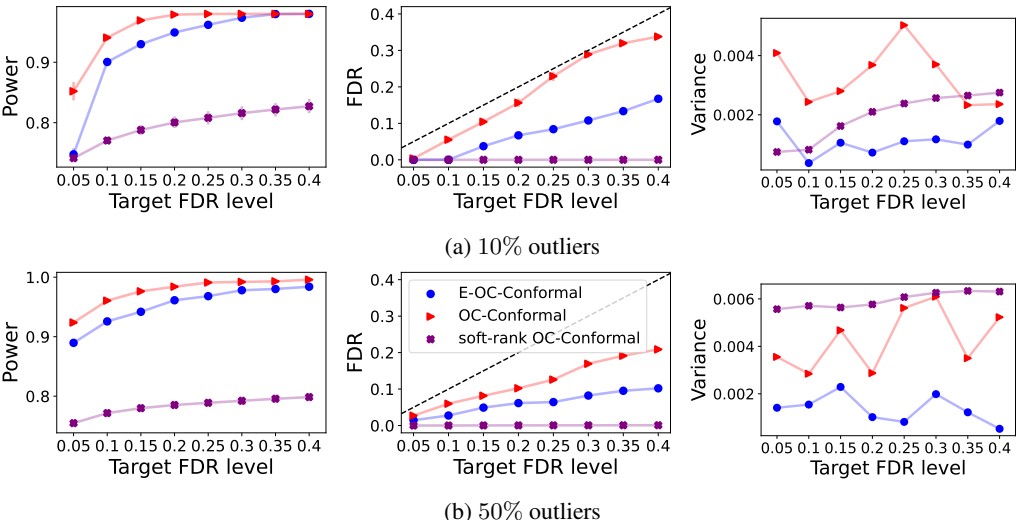

(a) 10% outliers

(b) 50% outliers

Figure S15: Performance on synthetic data of the proposed derandomized outlier detection method, `E-OC-Conformal`, applied with $K = 10$, compared to that of its randomized benchmark, `OC-Conformal`. These methods are also compared to the soft-rank method, applied with $K = 10$, `soft-rank OC-Conformal`, as a function of the target FDR level. All methods leverage a one-class support vector classifier. S15a presents the performance in high-power regime with signal amplitude $3.6$ when there are 10% outliers. S15b presents the performance in high-power regime with signal amplitude $3.4$ when there are 50% outliers. The dashed line indicates the nominal false discovery rate level. Note that these results correspond to 100 repeated experiments based on a single realization of the labeled and test data, hence why the results appear a little noisy.

powerful and less stable than `AdaDetect` on the studied data sets, and therefore we increase the number of analyses of our randomization procedure to $K = 70$.

Figure S17 indicates our martingale-based derandomization procedure indeed reduces the algorithmic variability while controlling the false discovery proportion. These results also demonstrate the trade-off between stability and power: the selections of `E-OC-Conformal` are more stable at the cost of having lower power compared to the base `OC-Conformal`. Focusing on the derandomization methods, when the test set contains 10% outliers (Figure S17a), the soft-rank method has a comparable and possibly slightly higher power compared to our martingale-based method. On the other hand, our method exhibits greater power for a larger proportion of outliers (Figure S17b). One explanation for this behavior is in our choice of $\alpha_{\mathrm{bh}} = 0.05$, which may not be optimal for situations with low proportions of outliers when using the one-class conformal algorithm. Observe also that for the KDDCup99 data set, when the test set contains 40% outliers, the selection variance of the soft-rank approach is the highest among the methods we study. One way to reduce this variance is to further increase the number of analyses $K$. We conclude this experiment with Figure S19, which provides a comprehensive comparison of the FDR and power for varying proportions of outliers in the test set. This figure confirms the reproducibility of our method—observe how the FDR, evaluated over 100 random sub-samples of the data, is controlled.

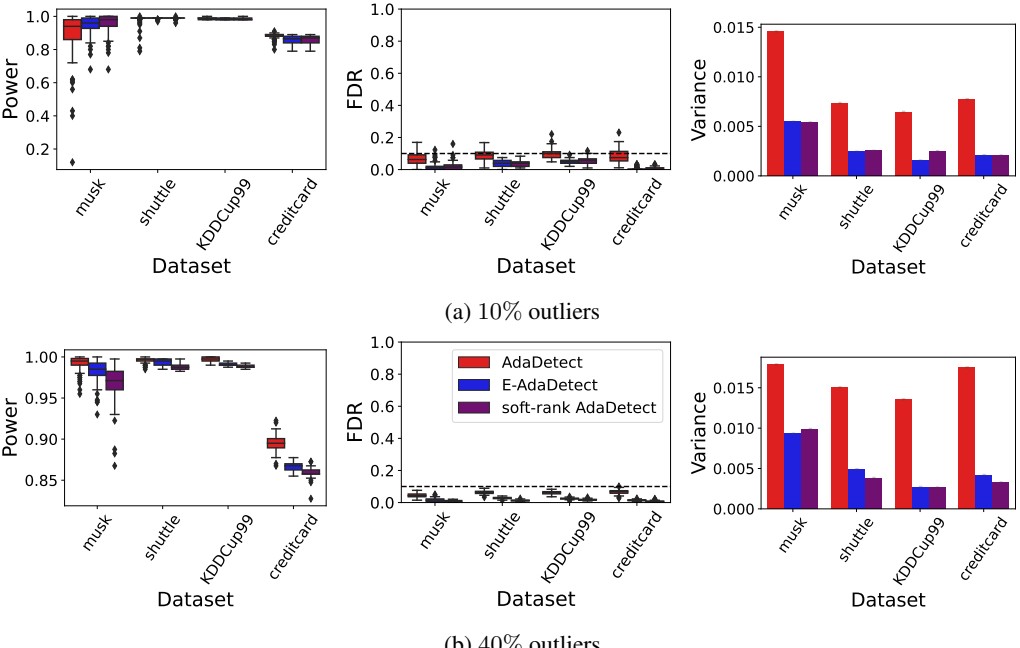

(a) 10% outliers

(b) 40% outliers

Figure S16: Performance on real data of `E-AdaDetect`, its randomized version, `AdaDetect`, and `soft-rank AdaDetect`. S16a and S16b present the performance of all methods for 10% and 40% outliers in the test-set, respectively. All methods leverage a random forest binary classifier. Left: average proportion of true outliers that are discovered (higher is better). Right: variability of the findings (lower is better).

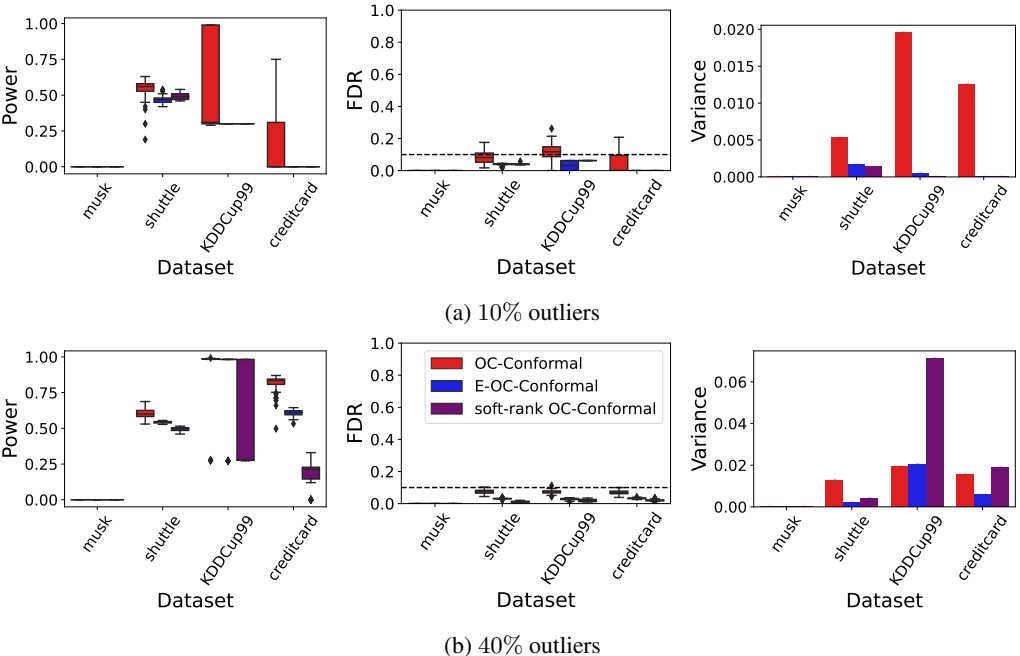

(a) 10% outliers

(b) 40% outliers

Figure S17: Performance on real data of `E-OC-Conformal`, its randomized version, `OC-Conformal`, and `soft-rank OC-Conformal`. S17a and S17b present the performance of all methods for 10% and 40% outliers in the test-set, respectively. All methods leverage an isolation forest classifier. Left: average proportion of true outliers that are discovered (higher is better). Right: variability of the findings (lower is better).

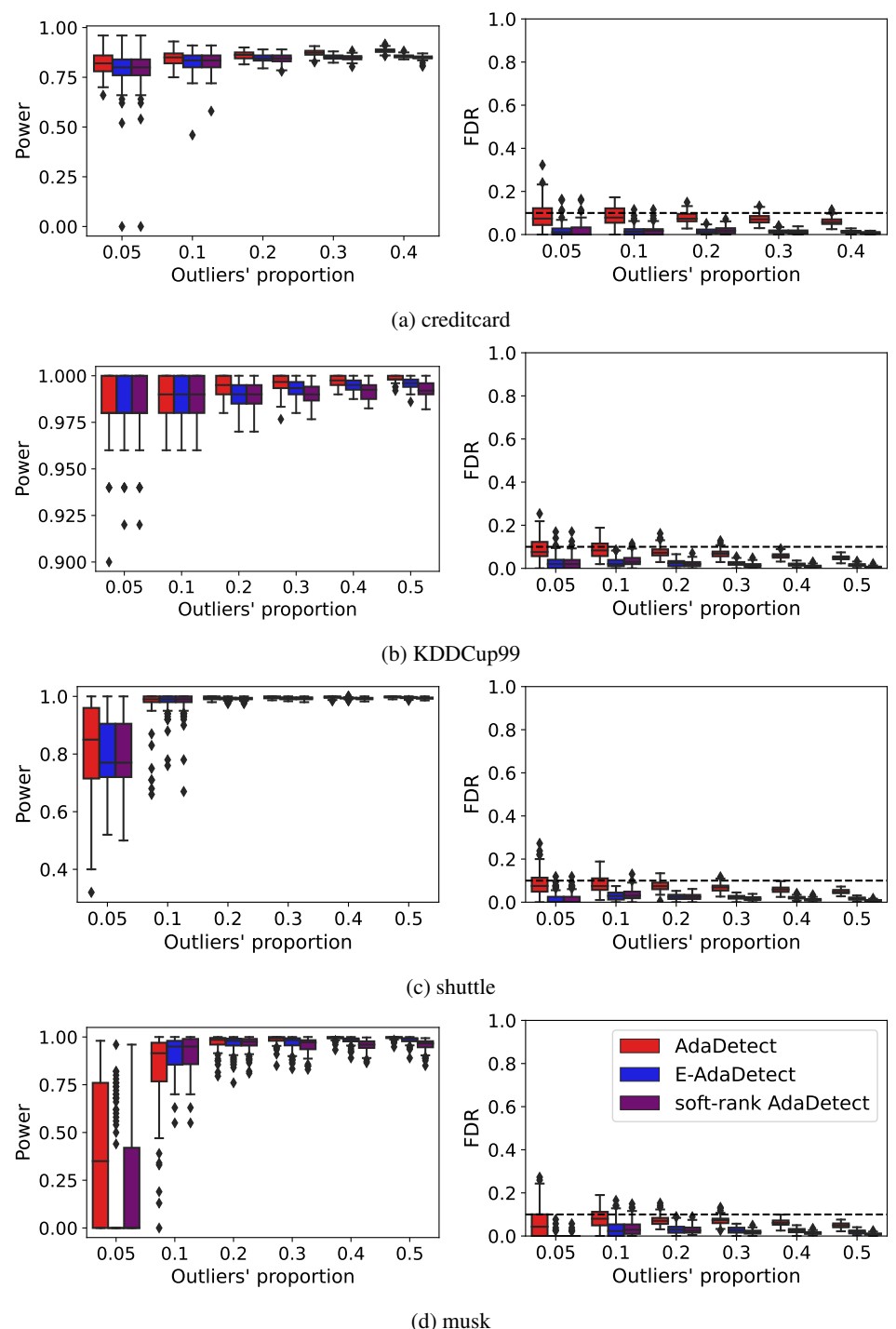

(a) creditcard

(b) KDDCup99

(c) shuttle

(d) musk

Figure S18: Performance on real data of `E-AdaDetect`, its randomized version, `AdaDetect`, and `soft-rank AdaDetect` as a function of the outliers proportion in the test-set. Each sub-figure corresponds to a different dataset. All methods leverage a random forest binary classifier. The results are averaged over 100 independent realizations of the data, which are randomly subsampled from the raw data sources. Other details are as in Figure S16.

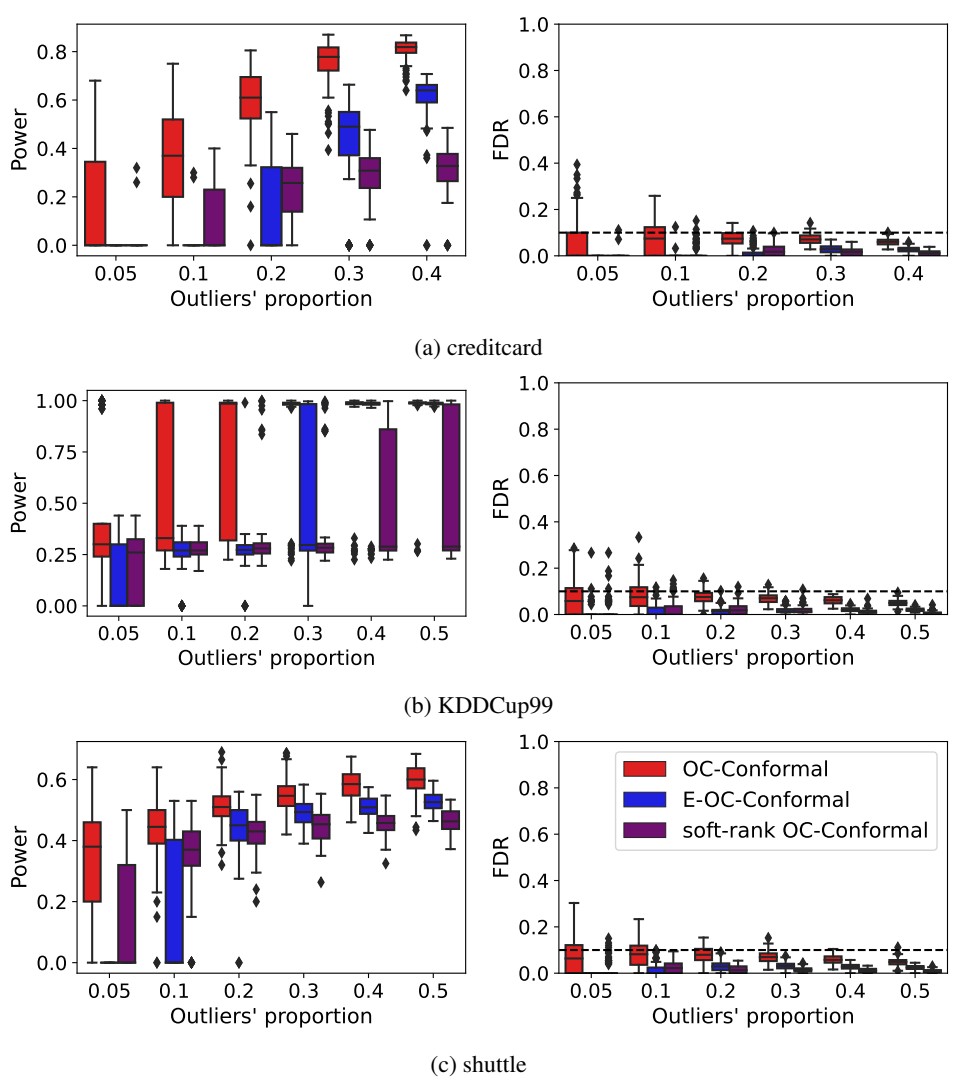

(a) creditcard

(b) KDDCup99

(c) shuttle

Figure S19: Performance on real data of `E-OC-Conformal`, its randomized version, `OC-Conformal`, and `soft-rank OC-Conformal` as a function of the outliers proportion in the test-set. Each subfigure corresponds to a different dataset. The power obtained for musk dataset is zero for all methods and thus is not shown. All methods leverage an isolation forest classifier. The results are averaged over 100 independent realizations of the data, which are randomly subsampled from the raw data sources. Other details are as in Figure S16.