# OpenReview forum: "Derandomized novelty detection with FDR control via conformal e-values"
_NeurIPS.cc/2023/Conference — NeurIPS 2023 poster_

### Official Review · Reviewer_ccpQ · 2023-07-03

**Soundness:** 3 good
**Presentation:** 3 good
**Contribution:** 3 good
**Rating:** 6
**Confidence:** 4

**Summary:**

This paper applied the derandomized e-value to the conformal novelty detection, which reduces the randomness of original approach using conformal p-value. The authors also refined the method by adaptively weighting the conformal p-values based on an estimate of the out-of-sample accuracy of each underlying machine learning model. Simulations with synthetic and real data are conducted to compare the performance with the original approach.

**Strengths:**

This paper is the first work to apply the derandomized e-value to the conformal inference, which makes it more stable compared with the original approach.

**Weaknesses:**

### 1. The novelty of this paper is limited.

Considering the previous works by applying derandomization and E-value to Knockoff filter [Ren et al., 2020, Ren and Baber 2023], it is straightforward to extend it to the conformal setting.

### 2. The challenges are not stated clearly.

 (1) From the connection between the conformal BH filter and Eq. (2) (given by Rava et al. [2021]), it would be easy to extend the construction in [Ren and Baber 2022] to conformal e-value in Eq. (5). In addition, the weighted e-value is also proposed by Ren and Baber [2022].

 (2) Given the Theorem 2 from Ren and Baber [2022], all the technical difficulty falls in proving Theorem 3.2. However, the proof strategy is from  Rava et al. [2021].

**Questions:**

1. What is the superiority of data-driven weights over the fixed width? Is the weighted approach in Ren and Baber [2022] applicable here?

2. In Figure 2, why is the power of E-AdaDetect higher than Ada-Detect while the FDR value is still lower? It seems contradictory to common sense.

**Limitations:**

The main limitation is the novelty. Despite that this is the first work to deploy derandomized e-value to split conformal inference, the framework and the theory is well studied in previous works. In addition, the technical contribution is minor.

---

> ### Author Rebuttal · Authors · 2023-08-04
>
> Thank you for your thoughtful review and constructive feedback. We are addressing your comments below.
>
> ### Novelty and data-driven weights
> Our data-adaptive weighting method involves technical innovations and is different from the use of “side-information” weights discussed in Ren and Baber [2022]. The key distinction is that our method weights the conformal e-values in a data-driven way, leveraging additional information extracted from the same data used to compute the conformal e-values themselves. This is challenging because standard methods for weighted hypothesis testing typically assume the weights to be independent, obtained from separate data, and it is not clear where such independent information could come from in our case. For example, Ren and Baber [2022] discuss how to leverage prior information obtained from a completely independent source in the context of the knockoff filter. Their approach is thus not applicable here.
>
> Finally, our experiments show that our data-driven weighting method leads to higher power (e.g., refer to Figure 4), and we believe this method could also be useful beyond the scope of this paper.
>
> ### Novelty and de-randomization in conformal inference
> The novelty of this work is enhanced by the fact that it is the first paper to specifically address the important problem of de-randomizing conformal inferences. We think this research can have broad practical impacts due to the rapidly spreading adoption of conformal inference methods and the well-known importance of mitigating algorithmic randomness in sensitive applications.
>
> ### Novelty and relation to Ren and Baber [2022] and Rava et al. [2021]
> Although our methods and proofs involve some technical similarities (which we clearly acknowledged), the problem studied in this paper is completely different from both that of Ren and Baber [2022] and that of Rava et al. [2021]. Ren and Baber [2022] focus on de-randomizing conditional independence tests based on knockoffs; they do not study conformal inference. Rava et al. [2021] study a problem closely related to conformal classification but they do not tackle de-randomization at all.
>
> ### Figure 2
> Finally, in Figure 2, it should not be surprising that the power of E-AdaDetect can sometimes be higher than that of Ada-Detect while the FDR is lower. The two methods may often produce completely different orderings of the test points. Therefore, it is perfectly plausible that one method sometimes provides better separation between the inliers and the outliers. There would only be an intuitive contradiction here if the two methods constructed their inferences by applying a different significance threshold to the same test statistics; but that is clearly not the case.

---

> > ### Comment · Reviewer_ccpQ · 2023-08-16
> > **Response to rebuttal**
> >
> > Thank you for answering my questions. As you said, data-adaptive weighting method is the main innovation of this paper. However, the corresponding results are not sufficient. I think the technical difficulty of data-driven weights is addressed by the symmetric assumption (line 206). Can you extend the results to more general cases? Also, there is no experiments on real data to show the superiority of data-driven weights.

---

> > > ### Author Response · Authors · 2023-08-17
> > > **Re: results of additional experiments based on real data**
> > >
> > > Thank you for the suggestion of including additional experiments to demonstrate the performance of our proposed data-driven weighting method on real data. Following your suggestion, we have carried out additional experiments using the same 4 real data sets considered in the paper: "musk", "shuttle", "KDDCup99", and "creditcard". We have uploaded the results, along with a detailed description of the setup, in this one-page blinded PDF file: https://docdro.id/8QEqtk8
> > >
> > > In summary, the results demonstrate clearly the advantage of the two alternative weighting schemes ("t-test" or "avg. score"),
> > > which in most cases lead to noticeably higher power compared to the benchmark with uniform weighting. Notably,
> > > we see that weighting based on the t-test is the most powerful approach. These results are consistent with those presented in Figure
> > > 4 of the paper, which were based on synthetic data. Further, it is interesting to note from Figure 1 (in the new one-page PDF) that data-driven
> > > weighting is also effective at further reducing the algorithmic randomness of our findings, leading to lower variance.
> > >
> > > We would of course be happy to include these results in the revised manuscript, possibly along with other similar results which we did not include in the one-page PDF for brevity.
> > >
> > > Finally, regarding your suggestion of extending our data-driven weighting method to more general cases (e.g., classification or regression, instead of outlier detection), this is certainly a good idea for future work. However, we feel that it would go beyond the scope of this paper to investigate methods for problems other than outlier detection, partly also due to to space limitations.

---

> > > > ### Comment · Reviewer_ccpQ · 2023-08-19
> > > >
> > > > Thanks for your response and additional experiments. I have updated my rating from 5 to 6.

---

### Official Review · Reviewer_TGb1 · 2023-07-06

**Soundness:** 2 fair
**Presentation:** 2 fair
**Contribution:** 3 good
**Rating:** 5
**Confidence:** 2

**Summary:**

The paper employs conformal e-values, as opposed to p-values, to quantify statistical significance during outlier testing under FDR control. This approach enables the principled aggregation of results from mutually dependent tests, thereby providing a solution to de-randomize (split) conformal inferences.

**Strengths:**

The paper addresses a significant issue of the problem of the randomness in conformal inferences.

They propose a method to make conformal inferences more stable by leveraging suitable conformal e-values instead of p-values to quantify statistical significance, also merging the idea of de-randomizing conformal novelty detection and FDR control.

The proposed method has the potential to significantly improve the stability and interpretability of conformal inferences.

**Weaknesses:**

The paper could significantly benefit from revisions aimed at improving clarity. The current presentation of ideas and concepts is convoluted, making it difficult for readers to follow and understand the arguments and methodologies proposed.

While the authors' approach is novel, it builds upon existing studies and techniques. The authors could strengthen the originality of their work by further highlighting the unique aspects of their approach and how it differs from previous methods.

The authors provide simulations to demonstrate the effectiveness of their method, but it would be beneficial to see more empirical evaluations, including comparisons with other state-of-the-art methods.

**Questions:**

The paper focuses on derandomizing split-conformal inferences. I wonder if jackknife+ would inherently solve the problem.

The authors mention that their method can be extended to leverage adaptive weights based on the data. Could they discuss potential strategies for choosing these weights in practice?

**Limitations:**

See weakness.

---

> ### Author Rebuttal · Authors · 2023-08-04
>
> Thank you for your review; we appreciate the effort and honest feedback. We are sorry to hear you found the paper a bit hard to understand, but we hope we can answer your questions here.
>
> - Clarity. Other reviewers found the paper to be clear, but we can try to make it even more accessible. It would help us if you could be a little more specific about which sections or concepts you found confusing.
>
> - Jackknife+. Section 1.3 explains the relation of our work with Cross-Validation+, of which the Jackknife+ is a special case. In short, the Jackknife+ is not a satisfactory solution for the problem considered in this paper due to the difficulty of controlling the FDR. However, in the future it may be possible to combine our work with the Jackknife+, as we suggested in Section 5.
>
> - Data-driven weights. The extension of our method based on data-driven weights is explained in Section 3.2. This also includes a detailed description of the specific approach adopted in the experiments of this paper, presented in Section 4.2.3. See also Algorithm S5 in the Supplementary Material for further details. Is it possible that you read this part of the paper quickly or missed the Supplementary Material?
>
> - Novelty. Our paper makes two new contributions. First, it is the first one to carefully study the problem of de-randomizing conformal inferences using e-values, while controlling the FDR. We think this will have direct practical impact and is also likely to open new directions of research. Second, our paper combines and innovates upon ideas from different fields, introducing a key technical novelty that allows power boosting through data-driven e-value weighting. We think this adaptive weighting strategy will be useful beyond the scope of this paper. In summary, it is true that our results are achieved by building upon the works of others, but the importance of their prior contributions is clearly acknowledged.
>
> - Simulations. The paper presents extensive simulations and empirical comparisons with state-of-the-art methods. Is it possible that you missed some of the details described in Section 4 and in the Supplementary Material? Or is there anything specific that you would like to see added?

---

### Official Review · Reviewer_Ed41 · 2023-07-07

**Soundness:** 3 good
**Presentation:** 3 good
**Contribution:** 3 good
**Rating:** 7
**Confidence:** 4

**Summary:**

The main limitation of conformal prediction lies in its inherent randomness. However, this paper presents an innovative solution by introducing a derandomized version of conformal prediction, specifically applied to the field of novelty detection. Through the incorporation of conformal e-values, the proposed method successfully reduces the element of randomness while providing provable and effective control over the False Discovery Rate (FDR).

The key contribution of this research lies in its pioneering use of e-values, instead of traditional p-values, within the framework of conformal prediction. This innovative approach significantly simplifies the aggregation process, reducing randomness without compromising the overall detection power.

**Strengths:**

This paper is a highly innovative and inspiring work that highlights the potential of e-values as a superior alternative to p-values for derandomizing conformal prediction through the aggregation of multiple dependent tests of the same hypothesis.

The paper introduces a novel approach for constructing e-values and provides a rigorous guarantee of false discovery rate (FDR) control, with Theorem 3.2 being the main contribution of the research.

Furthermore, the practical aspect of the paper lies in the deployment of conformal e-values in AdaDetect, which presents a derandomized version of AdaDetect. It is important to note that E-AdaDetect is not merely a simple combination but rather a specific practical application demonstrating how to aggregate e-values using data-adaptive weights.

The experimental results showcased in the paper reveal that the derandomized AdaDetect exhibits comparable power to its randomized counterpart, while effectively controlling the FDR. This finding is particularly surprising (even higher power? )and highlights the potential benefits of adopting the "new"(not same as the classical definition) e-values .

**Weaknesses:**

The main paper primarily focuses on experimental results obtained from synthetic data, which is sampled from simple Gaussians. It may be considered relatively easier compared to real-world scenarios. Therefore, I would suggest including at least one experiment using more complex synthetic data or real data from the supplementary material.

**Questions:**

*

**Limitations:**

*

---

> ### Author Rebuttal · Authors · 2023-08-04
>
> Thank you for your thoughtful review and constructive feedback.
>
> We also appreciate your suggestion of including an additional experiment utilizing more complex synthetic data or real data; if space permits we will insert these results in the revised manuscript.

---

### Official Review · Reviewer_gnkM · 2023-07-08

**Soundness:** 4 excellent
**Presentation:** 4 excellent
**Contribution:** 3 good
**Rating:** 7
**Confidence:** 4

**Summary:**

This papers proposes a way to reduce the randomness in novelty detection methods (detecting out-of-distribution points) that are based on the split-conformal inference paradigm. This is done by ensembling over several ($K$) train-validation splits of the dataset. The main technical point is to aggregate the evidence from the $K$ individual predictions by averaging their E-values, as a replacement to considering the p-value in the traditional approach with only one train-validation split. It is shown that the FDR (false discovery rate) can still be controlled with these quantities. The method is further enhanced by using certain weighted averages of the E-values, taking into account the estimated powers of each one. The methods are evaluated on synthetic and real data, showing that the proposed derandomized method works better if the fraction of outliers is larger.

**Strengths:**

The proposed novelty detection approach is original in that it combines evidences from different algorithm runs into a single evidence score. This required a formulation of the algorithms' evidences in terms of E-values instead of p-values that are commonplace for conformal methods, to avoid a loss in detection power. While prior novelty detection works have touched upon E-values, the current paper succeeds in controlling the FDR based on these E-value evidences. The methodological details (Sec. 3.1) and mathematical proof (Theorems 3.2 and 3.6) of this FDR control are strongly inspired by existing works, but this technique is new in the area of novelty detection.

As to significance, the method does reduce the variance as promised and keeps its FDR guarantee. It has higher power than competitor methods in some regimes, but quite consistently weaker power in other regimes, particularly for low fraction of outliers. Secondly, while the problem and the solution is built around quite specific requirements, it is plausible that the presented techniques can be applied to the derandomization of other inference methods with statistical guarantees, as the authors say in their Conclusion.

The clarity of the paper is exceptional (but see my comments on Appendix S2 below), the explanations and discussions are to the point.

**Weaknesses:**

As the biggest weakness of the paper I see that, despite the novelty of the approach, the considered problem requirements are relatively specific and the solution therefore narrow, as it concerns the de-randomization of certain novelty detection algorithms that are based on conformal inference and that aim at a mathematical control of the FDR.

Also, the method's performance (power) falls behind other methods in certain application regimes. It would be good to have some rules beforehand to know in which regime to apply which method.

If space permits, I would like to see some real data experiments (see Sec. 4.3) in the main text rather than only in the Supplement.

**Questions:**

* Please mention what K is in Fig. 1.
* line 228: Should this be D^(k)_cal \cup D_test?
* How is AdaDetect different from E-AdaDetect with K=1 (Fig. 3)?
* Please carefully re-write the Proof of Theorem 3.6 in Appendix S2, as I believe there are many typos and several things that could be explained better. In particular:
   - Please explain the first equality after line 35.
   - It seems that the martingale runs from l'=l down to l'=0, correct?
   - It would be good to explain in detail, over which random variables the various expectations E after line 26 run and which random variables are held fixed (conditioned on).
   - Right-hand-sides of Eq. (S1) and (S2): Should there be t instead of \hat{t}^{(k)}?
   - line 31: The word "outliers" should be "inliers"
   - Should the inf be changed to sup, or maybe the inequality sign reversed?
   - line 34: inside the curly brackets, the l should be l'?

**Limitations:**

Yes, this is discussed in a fair way after line 351.

---

> ### Author Rebuttal · Authors · 2023-08-04
>
> Thank you for your thoughtful review and constructive feedback.
>
> ### Broader relevance of our work
>
> It is true that this work focuses explicitly on de-randomizing conformal inferences for novelty detection tasks, which may seem like a relatively narrow scope compared to the broader range of possible conformal inference applications. However, the novelty detection idea of testing the exchangeability between a calibration set and a new test point is essentially the foundation of all conformal inference.
>
> This is why it makes sense to start tackling de-randomization in conformal inference from the perspective of novelty detection, but it is not hard to see how our ideas could be extended to other tasks, including regression and classification (after replacing the FDR with analogous concepts such as the FCR). Such extensions are suggested in Section 5. Space limitations prevent us from developing these extensions fully within the present paper, but this is certainly an interesting direction for follow-up work.
>
> ### The effect of de-randomization on power
>
> The goal of our method is to decrease the algorithmic randomness of existing conformal inference techniques, not to increase their power.
>
> Our results consistently demonstrate the effectiveness of our approach in achieving this goal (e.g., refer to Section 4).
> Reducing algorithmic randomness is important because it enhances the reproducibility, stability, and interpretability of any findings. There is no clear reason why de-randomization should also lead to higher power in general; it is a nice surprise that it sometimes does. However, it must be remembered that power may not always be the most meaningful concept in applications with high algorithmic randomness, as it refers to an expected behavior averaged over many hypothetical repeated experiments. In reality, algorithmic randomness can make the results of any analysis highly unpredictable, regardless of what the average power might be, and this is precisely the issue that motivates our work.
>
> Regarding your suggestion of providing some guidelines for practitioners on when to expect our method to also boost power, we would like to point out that we have already touched upon this issue in the paper. Specifically, we have explained how our method tends to reduce power in applications with few expected discoveries (e.g., refer to Section 5). Unfortunately, it might be challenging to predict power improvements more precisely without much stronger assumptions than those made in this paper.
>
> ### Other questions
>
> In Figure 3, the difference between AdaDetect and E-AdaDetect is that these two methods utilize different decision rules. The former computes p-values, while the latter relies on E-values. E-values are generally less powerful than p-values, but they have the advantage of facilitating de-randomization. This is why E-AdaDetect is at a disadvantage when applied with K=1, but this special case is not practically interesting (as it does not allow de-randomization) and it is only shown for completeness.
>
> We also thank you for bringing some typos and imprecisions to our attention; we will certainly fix them in the revised version of the paper.

---

> > ### Comment · Reviewer_gnkM · 2023-08-18
> >
> > Thank you for addressing my questions.
> >
> > I appreciate that you highlight the exchangeability as the central part of conformal inference, and that this comes out particularly in novelty detection. I am looking forward to seeing your approach being extended to derandomize other conformal tasks.
> >
> > Thanks also for your insights on what to expect on power. With this in mind, it is even surprising that your derandomized method improves power consistently in some regimes.
> >
> > I still encourage you to take some real-data experiments into the main paper, potentially even some of the ones you prepared as a response to Reviewer ccpQ during the rebuttal.
> >
> > I am raising my score to 7 and am recommending acceptance of the paper.

---

### Decision · Program_Chairs · 2023-09-21

**Decision:**

Accept (poster)

**Comment:**

This paper addresses a derandomized conformal e-value for outlier detection. It also introduces a data-adaptive weighting method where conformal e-values are weighted in a data-driven way, leveraging additional information extracted from the same data used to compute the conformal e-values themselves. The outlier detection method proposed in this paper seems to be original, although there are some concerns on its limited novelty. The authors did a good job in their rebuttal, resulting in an increase in the overall score. All reviewers feel that the paper has interesting contributions and is deserved for publication.